# Neural Differentiation Potential of Mesenchymal Stem Cells Enhanced by Biocompatible Chitosan-Gold Nanocomposites

**DOI:** 10.3390/cells11121861

**Published:** 2022-06-07

**Authors:** Huey-Shan Hung, Yi-Chin Yang, Chih-Hsuan Chang, Kai-Bo Chang, Chiung-Chyi Shen, Chien-Lun Tang, Szu-Yuan Liu, Chung-Hsin Lee, Chun-Ming Yen, Meng-Yin Yang

**Affiliations:** 1Graduate Institute of Biomedical Science, China Medical University, Taichung 40402, Taiwan; hunghs@mail.cmu.edu.tw (H.-S.H.); qsrgukp2010@gmail.com (C.-H.C.); kbwork2021@gmail.com (K.-B.C.); 2Translational Medicine Research Center, China Medical University Hospital, Taichung 40402, Taiwan; 3Department of Neurosurgery, Neurological Institute, Taichung Veterans General Hospital, Taichung 407204, Taiwan; jean1007@gmail.com (Y.-C.Y.); shengeorge@yahoo.com (C.-C.S.); chienluntang@gmail.com (C.-L.T.); syliu@vghtc.gov.tw (S.-Y.L.); zacklee228@gmail.com (C.-H.L.); chunmingyen@gmail.com (C.-M.Y.); 4Basic Medical Education Center, Central Taiwan University of Science and Technology, Taichung 406053, Taiwan; 5Graduate Institute of Medical Sciences, National Defense Medical Center, Taipei 11490, Taiwan; 6College of Medicine, National Chung Hsing University, Taichung 40227, Taiwan

**Keywords:** chitosan, gold nanoparticles, mesenchymal stem cell, neural differentiation, tissue engineering

## Abstract

Chitosan (Chi) is a natural polymer that has been demonstrated to have potential as a promoter of neural regeneration. In this study, Chi was prepared with various amounts (25, 50, and 100 ppm) of gold (Au) nanoparticles for use in in vitro and in vivo assessments. Each as-prepared material was first characterized by UV-visible spectroscopy (UV-Vis), Fourier-transform infrared spectroscopy (FTIR), atomic force microscopy (AFM), scanning electron microscopy (SEM), and Dynamic Light Scattering (DLS). Through the in vitro experiments, Chi combined with 50 ppm of Au nanoparticles demonstrated better biocompatibility. The platelet activation, monocyte conversion, and intracellular ROS generation was remarkably decreased by Chi–Au 50 pm treatment. Furthermore, Chi–Au 50 ppm could facilitate colony formation and strengthen matrix metalloproteinase (MMP) activation in mesenchymal stem cells (MSCs). The lower expression of CD44 in Chi–Au 50 ppm treatment demonstrated that the nanocomposites could enhance the MSCs undergoing differentiation. Chi–Au 50 ppm was discovered to significantly induce the expression of GFAP, β-Tubulin, and nestin protein in MSCs for neural differentiation, which was verified by real-time PCR analysis and immunostaining assays. Additionally, a rat model involving subcutaneous implantation was used to evaluate the superior anti-inflammatory and endothelialization abilities of a Chi–Au 50 ppm treatment. Capsule formation and collagen deposition were decreased. The CD86 expression (M1 macrophage polarization) and leukocyte filtration (CD45) were remarkably reduced as well. In summary, a Chi polymer combined with 50 ppm of Au nanoparticles was proven to enhance the neural differentiation of MSCs and showed potential as a biosafe nanomaterial for neural tissue engineering.

## 1. Introduction

Tissue engineering has wide-ranging applications in various fields, including cytology, biochemistry, molecular biology, biomedical engineering, and biomaterial science [1]. The demand for tissue engineering in biomedicine has gradually increased in recent decades. Tissue engineering has been used in various clinical applications, for example, vascular scaffolds [2], skin repair [3], bone tissue reconstruction [4], cartilage replacement [5], and cardiac muscle tissue regeneration [6]. In brief, the biomaterials that are applied in tissue regeneration approaches can be classified into natural and artificial types [7]. However, there are a number of factors that should be taken into consideration, such as biocompatibility and biodegradability [8]. A combination of artificial and natural biomaterials has been employed to develop novel applications capable of achieving better therapeutic effects [9]. Furthermore, nanotechnology has been applied to improve the therapeutic efficiency of biomedical approaches [10]. Nanoscale modification of biomaterials has been verified to exhibit superior abilities in various systems, including regenerative nanomedicines [11] or anti-cancer drug delivery devices [12], for clinical treatments.

Extracellular matrices (ECMs) such as fibronectin [13] and collagen [14] are now widely applied in tissue regenerative medicine. ECMs have been developed as scaffolds which can be fabricated with bioactive molecules [15] or nanoparticles [16] to further induce cell proliferation and differentiation. Chitosan (Chi) has been proven to exhibit good biocompatibility, biodegradability, and non-toxicity properties that are well-suited for biomedical applications [17]. Chitosan is a natural polymer that is purified from chitin, a compound found in abundance in crustaceans [18]. Chitosan is obtained from chitin, a polymer composed of randomly β-(1-4)-linked d-glucosamine and *N*-acetyl-d-glucosamine, by the process of deacetylation [19]. The water solubility of Chi can be influenced by the percentage of deacetylation [20]. Chi has many applications in current biomedical science, such as protein drug modification, polymer coating modification, and anti-adhesive capabilities [21]. Furthermore, Chi possesses good biocompatibility, biodegradability, and low toxicity, and therefore it can be applied in clinical approaches, including artificial skin/wound dressings and surgical sutures [22]. Chi has also been verified to induce the differentiation of stem cells into neuronal cells. However, the adherence of cells onto the Chi membrane has been shown to be poor [23]. After modification at nanoscale, Chi can be efficiently used as a drug delivery system and for cell differentiation-induced substrates [24].

Nanoparticles such as nanogold (denoted as Au) exhibit good biocompatibility, high stability, and non-reactivity with other molecules. Au nanoparticles can be modified with various materials to achieve novel biofunctions, such as drug delivery nanocarriers, bioimaging, or nanocomposite films for tissue engineering [25]. The manufacturing process of Au can be classified into two different processes: chemical manufacturing process and physical manufacturing process. Generally, in most previous studies, experiments were conducted using chemically manufactured Au. The chemical manufacturing process by which Au nanoparticles are synthesized is based on the reduction of a chloroauric acid (HAuCl_4_) solution with sodium citrate (Na_3_C_6_H_5_O_7_, working as a reducing agent) [26]. The size of the synthesized Au nanoparticles was approximately 5–20 nm. Additionally, Au nanoparticles can be obtained from physical manufacturing processes, such as the high temperature and pressure process, the crystal stacking method, and the epitaxial growth process [27]. Furthermore, physical vapor deposition (PVD) is an innovative high-tech physical method of obtaining edible Au nanoparticles [28]. The purity of PVD-processed Au nanoparticles can reach 99.99%, and the size of Au can be controlled between 0.5 and 50 nm. Compared with chemically manufactured Au, PVD-processed Au nanoparticles exhibited better biocompatibility and lower toxicity and had the additional advantage of not requiring the use of chemical pollutants [29]. Au obtained from the chemical reduction method demonstrated higher cytotoxicity and had other undesirable side effects due to the chemical residue on the Au nanoparticles [30]. Therefore, in our study, we prepared physical Au nanoparticles obtained by the PVD manufacturing process for Chi modification in our investigation of biocompatibility using in vitro assessments.

Stem cells have the characteristics of self-renewal and superior differentiation, which are important for tissue regeneration [31]. The stem cell treatments used for clinical neurodegenerative diseases include embryonic stem cells (ES cells), neural stem cells (NSCs), and mesenchymal stem cells (MSCs) [32]. MSCs are abundant in the human body with the majority present in bone marrow, which can differentiate into various cell types with multifunctionality [33]. Furthermore, MSCs can also be obtained from adipose tissue [34], muscle [35], dental pulp [36], and umbilical blood [37]. According to previous research, MSCs induce the lowest rate of immunological rejection in clinical applications [38,39]. MSCs were demonstrated to be an efficient cell resource in therapies for neurological disorders [40]. The combination of MSCs and suitable biomedical substrates can strengthen tissue regeneration by secreting various types of growth factors and cytokines [41]. The morphology of MSCs can manifest as axonal, neuron-like, and dendritic-like with the expression of neuronal differentiation markers (nestin and β-tubulin) [42]. Therefore, MSCs were used in in vitro assessments in our research.

In this study, the Chi polymer substrate was modified with various concentrations (25, 50, or 100 ppm) of physically manufactured Au nanoparticles (denoted as Chi–Au) to further investigate the physicochemical properties, biocompatibility, and induction of MSCs differentiation capacities. The ideal biomedical nanomaterials should provide a good microenvironment for biological functions and inhibit inflammatory responses by, for instance, decreasing the platelet–monocyte activation ratio. The expression of the neuronal differentiation markers, GFAP, β-tubulin, and nestin, were examined using in vitro assays. The endothelialization markers were determined by the expression of CD31 and the von Willebrand factor (vWF). Additionally, animal models involving implantation of Chi–Au nanocomposites were used to measure the biocompatibility of these compounds and to assess their potential efficiency in clinical tissue regeneration.

## 2. Materials and Methods

### 2.1. Material Preparation

#### 2.1.1. Chitosan Solution (Chi) Preparation

Acetic acid (100%) was purchased from SHIMAKYU CHEMICAL, Ltd. (Osaka, Japan). The chitosan solution was prepared by mixing 0.6 mL of acetic acid (100%), 0.6 g of chitosan (Sigma-Aldrich, Burlington, MA, USA), and 38.8 mL of deionized water (ddH_2_O) for 8 h at room temperature (RT) to obtain 1.5% chitosan dissolved in 1.5% acetic acid solution.

#### 2.1.2. Chitosan–Nanogold Composites Preparation

Physically manufactured nanogold particles (Au) were obtained from Gold NanoTech, Inc. (Taipei, Taiwan). The size of Au was roughly 3–5 nm. The aqueous Au solution was cautiously filtered using a 0.22 μm filter, and therefore, the Au solution was considered to be sterile. The 1.5% Chi solution described above was mixed with various concentrations of Au solution (25, 50, or 100 ppm) to obtain the Chi–Au solutions. The mass conservation formula M1V1 = M2V2 (M: concentration of solution, V: volume of solution) was applied to calculate the mixing ratio.

The nanocomposite films were prepared using the pure Chi and different concentrations of Chi–Au solutions, as described above. Each solution was coated on 15 mm glass coverslips or cell culture plates so that the surface was completely covered. The residual solution was removed by washing with PBS solution. The nanocomposite films were successfully prepared for further experiments.

### 2.2. Characterization of Materials

#### 2.2.1. Ultraviolet-Visible Spectroscopy (UV-Vis)

The as-prepared materials were examined with a UV-Vis spectrophotometer (Helios Zeta, ThermoFisher, Pittsfield, MA, USA). The range of wavelength was set at 190–1100 nm. The quartz cuvette was first washed by ddH_2_O and wiped with mirror paper. The background absorbance was measured after adding ddH_2_O. While measuring the as-prepared samples, the quartz cuvette was cleaned with ddH_2_O to remove any residual solution after each measurement. After each measurement, the result was analyzed using OriginPro 8 software (Originlab Corporation, Northampton, MA, USA).

#### 2.2.2. Fourier-Transform Infrared Spectroscopy (FTIR)

The as-prepared nanocomposites were measured using a Fourier-transform IR spectrometer (Shimadzu Pretige-21, Kyoto, Japan) to determine the functional groups. In brief, 0.06g potassium bromide (KBr, Sigma-Aldrich, Burlington, MA, USA) was first added into 200 µL of the as-prepared material. The mixtures were further dried out and independently scanned 8 times. The spectrum was then measured from 500 cm^−1^ to 4000 cm^−1^ at a 2 cm^−1^ resolution, as described in a previous study [43].

#### 2.2.3. Atomic Force Microscopy (AFM)

The surface topography of each as-prepared nanocomposite was independently examined using atomic force microscopy (MFP-3D, Asylum Research, Santa Barbara, CA, USA) [43]. In brief, 100 μL of nanocomposite was dropped on a Si wafer and then left to dry out. Afterward, the Si cantilever (Olympus AC240TS) coupled with a 2.0 N/m spring constant was used for viewing the surface of materials. The topography of each of the as-prepared nanocomposites was obtained at 512 × 512 pixels resolution in AC mode. Moreover, the surface roughness value was measured using ImageJ software (Version 5.0, Media Cybernetics Inc., Rockville, MD, USA).

#### 2.2.4. Dynamic Light Scattering (DLS)

DLS assay was executed using a Zetasizer Nano ZS instrument (Malvern Panalytical Ltd., Malvern, Worcestershire, UK). The data were analyzed through the manufacturer’s software. The measurement was manipulated by a 532 nm light source coupled with a 90° scatter angle. One (1) mL of each sample was added into a cuvette with an optical path of 1 cm. The experiment was performed in triplicate.

#### 2.2.5. Scanning Electron Microscopy (SEM)

The shape of each of the as-prepared nanocomposites was explored with a JEOL JEM-5200 scanning electron microscope (JEOL Ltd., Tokyo, Japan). Briefly, 10 μL of the Au nanoparticle solution was added on a silicon wafer and dried out at 80 °C. After being dried out, the silicon wafer with Au nanoparticles was sputter coated with silver. Next, the voltage was set at 5.0 kV to observe the shape of Au nanoparticles. The scale bar was set at 1 μm. The size of each of the nanoparticles was further analyzed using Image Pro software (*n* = 10).

### 2.3. Assessments of Biocompatibility

#### 2.3.1. Culture of Mesenchymal Stem Cells (MSCs)

Human umbilical cord MSCs were isolated from Wharton’s jelly tissue. The MSCs were subjected into an H-DMEM culture medium (Invitrogen, Waltham, MA, USA) that was supplemented with 10% fetal bovine serum (FBS), 1% sodium pyruvate, and 1% (*v/v*) antibiotics (100 U/mL penicillin/streptomycin) at 37 °C, in an atmosphere containing 5% CO_2_. The MSCs used in our research were at the 8th passage.

The immunophenotypes of the mesenchymal stem cells were identified using flow cytometry. The cells were first detached by EDTA (2 mM) with PBS. Next, the MSCs were cautiously washed by PBS containing both 0.1% sodium azide (Sigma, USA) and 2% bovine serum albumin (BSA). Subsequently, the cells were cultured with specific antibodies. The antibodies were combined with fluorescein isothiocyanate (FITC), phycoerythrin (PE), and PerCP-Cy5-5-A, which were denoted as follows: CD14-FITC, CD34-FITC, CD45-FITC, CD44-PE, CD90-FITC, and CD105-PerCP-Cy5-5-A. The FITC/PE/PerCP-Cy5-5-A conjugated IgG1 were applied as isotype controls (BD Pharmingen, San Diego, CA, USA). Lastly, the MSCs were identified using a flow cytometer (LSR II, BD Pharmingen).

#### 2.3.2. Cell Colony Formation Assay

The as-prepared nanocomposites were coated onto 24-well culture plates. There were 5 groups of culture plates: TCPS (tissue culture plate), pure Chi, and Chi with 25, 50, or 100 ppm of Au. Afterward, the MSCs (1 × 10^5^ cells/well) were cultured in the 24-well culture plates. After 3, 5, and 7 days of incubation (37 °C, 5% CO_2_), the cell morphology was evaluated with an optical microscope. The cell colonies were further calculated using Image-Pro Plus (Media Cybernetics, USA).

#### 2.3.3. Platelet and Monocyte Activation Assessments

The human monocytes collected from whole blood were obtained from a healthy volunteer. The process followed was the Percoll protocol (Sigma-Aldrich, St. Louis, MO, USA), with institutional review board approval (CE12164) from Taichung Veteran Hospital. The cell concentration was adjusted to 1 × 10^5^ cells using a culture medium containing 1% (*v/v*) antibiotics (10,000 U ml/penicillin G and 10 mg ml/streptomycin) and 10% fetal bovine serum (FBS), and the monocytes were incubated in 24-well plates coated with the as-prepared materials for 96 hr at 37 °C with 5% CO_2_. The monocytes were harvested by 0.05% trypsin, and the monocytes’ converting ratio was observed by microscope. The inflammatory response was identified by immunofluorescence staining using CD68 (a marker of macrophages) as a biomarker of the anti-CD68 primary antibody (GeneTex Incorporation, Irvine, CA, USA).

Subsequently, 2 × 10^6^ platelets per well were cultured onto the as-prepared nanocomposites in an incubator (37 °C, 5% CO_2_) for 24 h. The platelets were then fixed in 2.5% glutaraldehyde solution for at least 8 h. Afterward, the platelets were rinsed two times by PBS and dehydrated using 30–100% alcohol. After drying out, the platelets’ morphology in various treatments was evaluated by JEM-5200 SEM (JEOL Ltd., Akishima, Japan).

#### 2.3.4. Examination of Reactive Oxygen Species (ROS)

A DCFH-dA (2′,7′-dichlorofluorescin diacetate) (Sigma-Aldrich, USA) fluorescent probe was used to target reactive oxygen species in the cells. We cultured 2 × 10^5^ MSCs per well in 6-well culture plates with the various as-prepared nanocomposite coatings and incubated for 48 h. Next, the MSCs were collected by 0.05% trypsin-EDTA and washed twice with PBS. After removal of the supernatant, 10 nM of DCFH-dA was used to target intracellular ROS in the dark (37 °C, 30 min). The ROS production was then identified using a flow cytometer. The data were analyzed by Flow Jo software (Version 7.6, Becton Dickinson, Canton, MA, USA).

### 2.4. Examination of Matrix Metalloproteinase Expression

We cultured 2 × 10^5^ MSCs per well onto 6-well culture plates coated with the as-prepared nanocomposites. The culture plates were incubated for 48 h, and then the cultured medium was collected for further measurements. Next, the proteins were separated by 10 % SDS-PAGE gel with 2% gelatin. After electrophoresis, the pH 8.5 reaction buffer (2.5% Triton X-100, 0.2 M NaCl, 10 mM CaCl_2_, and 40 mM Tris-HCl) was rinsed twice for 30 min at room temperature. Furthermore, the pH 8.5 development buffer (0.01% NaN3, 0.2 M NaCl, 10 mM CaCl_2_, and 40 mM Tris-HCl) was added to reach activation in a 37 °C water bath for 12 h. Coomassie Brilliant Blue solution (50% methanol, 10% acetic acid) was used to stain the gels. Next, the stained gels were washed by a destaining buffer (20% methanol and 10% acetic acid). The clear bands were the protease-digested area which could be found within the dark blue background. Lastly, the gels were scanned, and we quantified the protein expression using Gel-Pro Analyzer 4.0 (Media Cybernetics, USA).

### 2.5. Real-Time Polymerase Chain Reaction (PCR)

The RNA expression in MSCs was extracted by TRIzol (lnvitrogen, ThermoFisher Scientific, Waltham, MA, USA), and the experiments were processed following the manufacturer’s instructions. The 2 × 10^5^ MSCs per well were cultured in 6-well culture plates with the as-prepared nanocomposite coatings. The cells were incubated for 3, 5, and 7 days at 37 °C, 5% CO_2_. After incubation, the medium was removed, and TRIzol solution (1 mL) was added into each well for 5 min to collect the cells. Afterwards, 200 μL of chloroform (Sigma, USA) was added into each well for 15 s, and the culture plates were left for 3 min at RT. Next, the samples were collected for 12,000 rpm, 15 min, 4 °C centrifugation. Afterward, the supernatant was discarded, and isopropanol (500 μL) was loaded (4 °C, 10 min), followed by centrifugation for 15 min (12,000 rpm, 4 °C). The samples were washed two times with 75% alcohol (1 mL) after discarding the supernatant. The samples containing RNA were dried out. Next, 20 μL of DEPC-treated H_2_O was loaded into the samples, while the data were analyzed at 260 nm by a ELISA reader (SpectraMax M2, Molecular Devices, San Jose, CA, USA. RevertAid^TM^ First Strand cDNA Synthesis Kit (Fermentas, Burlington, ON, Canada) was applied for cDNA synthesis following the manufacturer’s instructions, as described in previous research [44]. The RNA expression in MSCs with various treatments was identified by Step One^TM^ Plus Real-Time PCR System.

### 2.6. Immunofluorescence (IF) and Fluorescence Activated Cell Sorting (FACS) Analysis

In brief, 1 × 10^4^ of MSCs were seeded onto coverslip glasses (15 mm) with as-prepared nanocomposite coatings in 24-well culture plates for 3, 5, and 7 days’ incubation. The MSCs were washed three times with PBS and blocked by 5% FBS after incubating with various primary antibodies: CD44, Vinculin, GFAP, Nestin, β-Tubulin, CD31, and vWF (1:250 dilution, Santa Cruz, TX, USA) at 4 °C for 8 h. Next, the samples were rinsed and incubated with 1:300 dilution FITC or PE-conjugated secondary antibodies (Santa Cruz, USA) for 1 h. A DAPI (4,6-diamidion-2-phenylindole) solution (Invitrogen, USA) was applied to locate nuclei. The sample was rinsed with 50% glycerol/PBS solution and sealed for further observation. Fluorescent images were acquired using a fluorescence microscope (ZEISS AXIO IMAGER A1, Carl Zeiss AG, Jena, Germany). Furthermore, the target cells were identified using ImageJ software (Version 5.0, Media Cybernetics, Burlington, MA, USA).

Additionally, CD44, GFAP, Nestin, and β-Tubulin fluorescein-positive cells were identified using a fluorescence-activated cell sorting (FACS) Calibur flow cytometer (BD Biosciences, USA). The data were analyzed with version 7.6.1 Flow J software. All experiments were displayed in triplicate.

### 2.7. In Vivo Assessments

Female Sprague Dawley (SD) rats that were 2–3 months old and weighed 300–350 g were used in our research, with the approval of the China Medical University Animal Care and Use Committee (102-83-N). A rat’s dorsal skin was incised at a depth of 10 mm in order to implant the as-prepared nanocomposites after local anesthesia. The wound tissue was resected after one month for further experiments. Fibrous capsules were observed at 6 sites by hematoxylin and eosin (H&E) staining, which was quantified by ImageJ 5.0 software. The collagen deposition was detected by Masson’s trichrome staining (Sigma, USA). Macrophage (M1 & M2) polarization was detected using 1:200 dilution of anti-mouse monoclonal CD86 and CD163 antibodies (Santa Cruz, USA). Next, immunohistochemical staining of CD31 expression was processed by 3-Amino-9-Ethylcarbazole (AEC) chromogen (ScyTek Laboratories Inc., Logan, UT, USA), following the manufacturer’s instructions. APC anti-mouse CD31 antibodies were applied to observe endothelialization. Moreover, the signal amplification was conducted through 1:500 dilution of AF488 Donkey anti-mouse IgG secondary antibodies (Invitrogen, USA). Furthermore, the samples were cultured with anti-mouse monoclonal CD45 antibodies (Santa Cruz, USA). The cells were detected using DAB (3,3’-diaminobenzidine) solution and counterstained with hematoxylin. An Olympus ix71 fluorescence microscope (Tokyo, Japan) was used to evaluate the fluorescence intensity, and a DAPI solution was applied to determine the location of cell nuclei. The results are exhibited as mean ± SD (*n* = 5).

### 2.8. Statistical Analysis

In our research, the data of each assessment (*n* = 3 to 6) were obtained to avoid uncertainty, and the results are exhibited as mean ± Standard Deviation (SD). Student’s *t*-test was applied to determine the differences between various treatments. The *p* value < 0.05 elucidated significant differences.

## 3. Results

### 3.1. Materials Characterization

A brief illustration of the steps involved in preparing Chi–Au nanocomposites is shown in Figure 1A. The as-prepared nanocomposites, i.e., pure Chi, Chi–Au 25 ppm, Chi–Au 50 ppm, and Chi–Au 100 ppm, were first subjected to characterizations of their physicochemical properties. Figure 1B (a) shows the Au nanoparticles observed by SEM. Figure 1B (b) depicts the size distribution of Au which was analyzed by DLS assay, and the Au diameter was quantified as 45 ± 3.2 nm (Figure 1B (c)). The UV-Vis absorption peak at 520 nm indicated there were Au nanoparticles in the as-prepared nanocomposites (Figure 1C). Furthermore, the functional groups of each as-prepared nanomaterial were identified by FTIR analysis (Figure 1D). The specific chemical bond of Chi was at 3441 cm^−1^, which was determined to be a -OH bond. After combining the Chi and Au nanoparticles, the intensity of the -OH bond in Chi–Au 25, Chi–Au 50, and Chi–Au 100 was shifted to 3641 cm^−1^, 3651 cm^−1^, and 3436 cm^−1^, respectively. Additionally, the peak at 1640–1540 cm^−1^ indicated the amide functional group in Chi and the peak of the amide band were changed after the addition of Au nanoparticles. The FTIR spectra are displayed in Figure 1D.

The surface topography of the as-prepared nanocomposites was observed by AFM, and the images are shown in Appendix A. Appendix A displays the appearance at 1 × 1 µm^2^, and Appendix A shows the appearance at 5 × 5 µm^2^. The values of the root mean square (RMS) were further analyzed and were determined to be 1.49 nm, 1.47 nm, 0.67 nm, and 0.38 nm.

### 3.2. Phenotype Characterization of MSCs

The MSCs used in our research were characterized by the specific surface markers, including CD14, CD34, CD 45, CD44, CD90, and CD105, that were detected using flow cytometry. The histograms and quantification data are shown in Appendix A. Based on the results, the expression of CD14, CD34, and CD45 markers were quantified as 1% (negative markers), which indicated the MSCs did not express endothelial differentiation markers. Additionally, large amounts of cells expressed CD44 (99%), CD90 (97%), and CD105 (96%) (positive markers), which are associated with human MSCs.

### 3.3. Colonization Capacity of MSCs

A colonization assessment to evaluate the colony formation capacity of MSCs stimulated by Chi–Au nanocomposites was conducted. The images at 1, 3, 5, and 7 days were observed using an optical microscope and displayed in Figure 2A. Afterward, the colony formation was calculated as shown in Figure 2B. In the control group, the colony formation showed a 1-fold increase at 1, 3, 5, and 7 days. In the pure Chi group, colony formation was greater than in the control, but there was no significant difference (Day 1: ~5.35-fold increase, Day 3: ~5.15-fold increase, Day 5: ~5.09-fold increase, and Day 7: ~4.99-fold increase). In the Chi–Au 25 ppm group, the colony formation amounts were remarkably higher than in the control group (Day 1: ~7.76-fold increase (** *p* < 0.01), Day 3: ~7.27-fold increase (** *p* < 0.01), Day 5: ~6.9-fold increase (** *p* < 0.01), and Day 7: ~6.56-fold increase (** *p* < 0.01)). In the treatment with Chi–Au 50 ppm, the colony numbers were significantly greater than in the control (Day 1: ~13.16-fold increase (*** *p* < 0.001), Day 3: ~11.59-fold increase (*** *p* < 0.001), Day 5: ~12.98-fold increase (*** *p* < 0.001), and Day 7: ~9.26-fold increase (** *p* < 0.01)). Additionally, the results of the Chi–Au 100 ppm group were superior to those of the other groups (Day 1: ~24.58-fold increase (*** *p* < 0.001), Day 3: ~22.41-fold increase (*** *p* < 0.001), Day 5: ~25.65-fold increase (*** *p* < 0.001), and Day 7: ~20.28-fold increase (*** *p* < 0.001)). According to the data, the Chi–Au nanocomposites were shown to be capable of remarkably inducing MSCs’ colonization.

The CD44 expression in MSC colonies at 7 days was further investigated by the fluorescence-activated cell sorting (FACS) method. The results based on the CD44 expressed intensity are depicted in Figure 2C. The expression of CD44 in the control group showed a 1-fold increase at 7 days. In the pure Chi group, the result was not significantly different compared with the control (Day 7: ~1.04-fold increase). The Chi–Au 25 ppm treatment induced lower CD44 expression in MSCs (Day 7: ~0.78-fold, (* *p* < 0.05)). In the Chi–Au 50 ppm group, the expression of CD44 was also significantly lower than in the control (Day 7: ~0.71-fold (** *p* < 0.01)). Furthermore, the CD44 expression in the Chi–Au 100 ppm group was not significantly different compared with the control (Day 7: ~0.86-fold). Based on the above evidence, Chi–Au nanocomposites could significantly enhance MSCs’ proliferation. Furthermore, the expression of CD44 was significantly lower in Chi–Au treatment, indicating that Chi–Au nanocomposites could induce MSCs to undergo differentiation.

### 3.4. Biocompatibility Assessments

The biocompatibility of the as-prepared Chi–Au nanocomposites was further investigated. The activation of platelets after various treatments for 24 h was observed by SEM. The images are displayed in Figure 3A. The active form of the platelets was flattened, which was found in the control group. Following the Chi–Au 50 ppm treatment, the morphology of platelets changed to a round shape, which is a non-active form. Additionally, the number of adhered platelets and degree of activation of platelets in each treatment were quantified, as shown in Figure 3C. The numbers of adhered platelets in the control, Chi, Chi–Au 25 ppm, Chi-Ai 50 ppm, and Chi–Au 100 ppm groups were 61 × 10^3^, 51 × 10^3^, 19 × 10^3^, 11 × 10^3^, and 18 × 10^3^ cells. The degree of activation at the range of 0.0–1.0 was determined, and the results were as follows: control 1.0, Chi 0.81 (* *p* < 0.05), Chi–Au 25 ppm 0.18 (*** *p* < 0.001), Chi–Au 50 ppm 0.062 (*** *p* < 0.001), and Chi–Au 100 ppm 0.375 (** *p* < 0.01).

The conversion of monocytes into macrophages was investigated. It is known that at certain quantifiable levels of the conversion of monocytes (~5 μm) into macrophages (~40 μm), an immune response is stimulated. Figure 3B shows the immunofluorescence images associated with CD68 (macrophage marker) expression in monocytes at 96 h. The semi-quantification results in Figure 3D demonstrate that the Chi–Au nanocomposite treatments inhibited CD68 expression in monocytes, among which the Chi–Au 50 ppm group had the lowest expression intensity (control 0.97, Chi 0.9, Chi–Au 25 ppm 1.36 (*** *p* < 0.001), Chi–Au 50 ppm 1.33 (*** *p* < 0.001), Chi–Au 100 ppm 1.86 (*** *p* < 0.001)). Furthermore, the monocyte conversion yield was also quantified in Appendix A. The number of monocytes was quantified as follows: control 67 × 10^4^, Chi 72 × 10^4^, Chi–Au 25 ppm 66 × 10^4^, Chi–Au 50 ppm 42 × 10^4^, and Chi–Au 100 ppm 52 × 10^4^ (Appendix A). The numbers of macrophages were calculated, as shown in Appendix A (control 21 × 10^4^, Chi 16 × 10^4^, Chi–Au 25 ppm 9 × 10^4^, Chi–Au 50 ppm 5 × 10^4^, and Chi–Au 100 ppm 11 × 10^4^). Additionally, the monocyte conversion yield was analyzed (Appendix A), and the results indicated Chi–Au 50 ppm treatment was the lowest (~0.38-fold increase, *** *p* < 0.001), followed by Chi–Au 25 ppm (~0.48-fold increase, ** *p* < 0.01), Chi–Au 100 ppm (~0.67-fold increase, * *p* < 0.05), Chi (~0.71-fold increase, * *p* < 0.05), and the control (1-fold increase).

The ROS production in MSCs stimulated by Chi–Au nanocomposites was examined at 48 h. The semi-quantification was evaluated as Figure 3E. The results indicate that the Chi–Au 50 ppm treatment stimulated the lowest ROS production in MSCs (control 1-fold increase, Chi ~0.52-fold increase (** *p* < 0.01), Chi–Au 25 ppm ~0.46-fold increase (*** *p* < 0.001), Chi–Au 50 ppm ~0.34-fold increase (*** *p* < 0.001), Chi–Au 100 ppm ~0.37-fold increase (*** *p* < 0.001)). The evidence elucidates that the Chi–Au 50 ppm treatment significantly inhibited platelet activation, decreased CD68 expression, reduced monocyte conversion, and decreased ROS generation, suggesting combining Chi with 50 ppm of Au nanoparticles produced a nanocomposite with superior biocompatibility for tissue regeneration.

### 3.5. Assessments of Matrix Metalloproteinase (MMP) Activity and Vinculin Expression

Figure 4 shows the amounts of MMP-2/9 expression in MSCs induced by various nanocomposite treatments at 48 h. The MMP-2/9 enzymatic activities are shown in Figure 4A. Afterward, the expression of MMP-9 in each group was semi-quantified, and the results were as follows: control 1-fold increase, Chi ~1.03-fold increase, Chi–Au 25 ppm ~0.93-fold increase, Chi–Au 50 ppm ~1.21-fold increase (* *p* < 0.05), and Chi–Au 100 ppm ~1.32-fold increase (* *p* < 0.05) (Figure 4B).

The vinculin expression in MSCs at 1, 3, 5, and 7 days was evaluated by immunofluorescence staining. The images were acquired by fluorescence microscope, as shown in Figure 4C. The semi-quantification results displayed in Figure 4D were as follows: control 1-fold increase at 1, 3, 5, and 7 days, Chi (Day 1: ~1.58-fold increase, Day 3: ~1.17-fold increase, Day 5: ~1.56-fold increase, Day 7: ~1.26-fold increase), Chi–Au 25 ppm (Day 1: ~2.53-fold increase (** *p* < 0.01), Day 3: ~2.44-fold increase (* *p* < 0.05), Day 5: ~3.14-fold increase (* *p* < 0.05), Day 7: ~2.18-fold increase (* *p* < 0.05)), Chi–Au 50 ppm (Day 1: ~2.14-fold increase (* *p* < 0.05), Day 3: ~2.2-fold increase (* *p* < 0.05), Day 5: ~3.83-fold increase (** *p* < 0.01), Day 7: ~1.53-fold increase), Chi–Au 100 ppm (Day 1: ~1.31-fold increase, Day 3: ~0.98-fold increase, Day 5: ~2.5-fold increase, Day 7: ~1.18-fold increase). Appendix A shows the influence of various nanocomposites on the morphology of MSCs, i.e., the aspect and diameter. The results exhibit that the combination of Chi and Au 50 ppm could enhance the MMP-9 and vinculin expression to strengthen tissue regeneration capacities.

### 3.6. Real-Time PCR Assay for mRNA Expression Induced by Chi–Au

To investigate the mRNA expression of differentiation markers in MSCs treated with various nanocomposites at 3, 5, and 7 days, the real-time PCR assay was applied for further detection. The differentiation markers included neural (GFAP, β-Tubulin, and nestin) and endothelial (CD31 and vWF) proteins. The quantification results are displayed in Figure 5. Figure 5A shows Chi–Au 50 ppm significantly enhanced the expression of neural-related proteins in MSCs. The data tables are demonstrated as Table 1, Table 2 and Table 3. Afterward, Figure 5B depicts the expression of the endothelial differentiation markers (CD31 and vWF) in MSCs. The data tables are exhibited as Table 4 and Table 5. The above results demonstrate that Chi–Au 50 ppm significantly facilitated the expression of neural differentiation proteins in MSCs.

### 3.7. FACS and IF Measurement for Protein Expression Induced by Chi–Au

The expression of neural and endothelial differentiation proteins in MSCs was also examined by the fluorescence-activated cell sorting (FACS) method (Figure 6A) and immunofluorescence (IF) staining (Figure 6B). The FACS quantification of the neural differentiation protein expression was displayed in Figure 6A. The Chi–Au 50 ppm nanocomposites were found to significantly enhance the expression of neural-related proteins in MSCs. The results are as follows: GFAP: (control (1-fold increase at 3, 5, and 7 days), Chi (Day 3: ~1.22-fold increase, Day 5: ~1.19-fold increase, Day 7: ~1.37-fold increase), Chi–Au 25 ppm (Day 3: ~1.50-fold increase (* *p* < 0.05), Day 5: ~1.64-fold increase (** *p* < 0.01), Day 7: ~2.18 fold (* *p* < 0.05)), Chi–Au 50 ppm (Day 3: ~1.72-fold increase (* *p* < 0.05), Day 5: ~2.16-fold increase (** *p* < 0.01), Day 7: ~2.56-fold increase (* *p* < 0.05)), and Chi–Au 100 ppm (Day 3: ~1.49-fold increase, Day 5: ~1.52-fold increase, Day 7: ~1.76-fold increase)); β-Tubulin: (control (1-fold increase at 3, 5, and 7 days), Chi (Day 3: ~1.09-fold increase, Day 5: ~1.13-fold increase, Day 7: ~1.28-fold increase), Chi–Au 25 ppm (Day 3: ~1.22-fold increase (* *p* < 0.05), Day 5: ~1.55-fold increase (* *p* < 0.05), Day 7: ~2.13-fold increase (* *p* < 0.05)), Chi–Au 50 ppm (Day 3: ~1.47-fold increase (* *p* < 0.05), Day 5: ~1.64-fold increase (* *p* < 0.05), Day 7: ~3.28-fold increase (* *p* < 0.05)), and Chi–Au 100 ppm (Day 3: ~1.27-fold increase, Day 5: ~1.42-fold increase, Day 7: ~2.01-fold increase)); nestin: (control (1-fold increase at 3, 5, and 7 days), Chi (Day 3: ~1.02-fold increase, Day 5: ~1.25-fold increase, Day 7: ~1.13-fold increase), Chi–Au 25 ppm (Day 3: ~1.26-fold increase (* *p* < 0.05), Day 5: ~1.65-fold increase (* *p* < 0.05), Day 7: ~2.41-fold increase (* *p* < 0.05)), Chi–Au 50 ppm (Day 3: ~1.39-fold increase (* *p* < 0.05), Day 5: ~2.42-fold increase (* *p* < 0.05), Day 7: ~2.77-fold increase (* *p* < 0.05)), and Chi–Au 100 ppm (Day 3: ~1.08-fold increase, Day 5: ~1.45-fold increase, Day 7: ~1.78-fold increase)) (Figure 6A).

The images of the endothelial differentiation protein expression at 7 days are demonstrated in Figure 6B. The images captured at 3 and 5 days are shown in Appendix A. The semi-quantitative results of the endothelial differentiation proteins’ expression based on fluorescence intensity are depicted in Figure 6C. Furthermore, in Figure 6C, the expression of endothelial differentiation proteins was quantified as follows: CD31: (control (1-fold at 3, 5, and 7 days), Chi (Day 3: ~1.33-fold increase (* *p* < 0.05), Day 5: ~1.03-fold increase, Day 7: ~1.22-fold increase), Chi–Au 25 ppm (Day 3: ~1.38-fold increase (** *p* < 0.01), Day 5: ~1.5-fold increase (* *p* < 0.05), Day 7: ~1.36-fold increase (** *p* < 0.01)), Chi–Au 50 ppm (Day 3: ~1.34-fold increase (* *p* < 0.05), Day 5: ~1.24-fold increase, Day 7: ~1.64-fold increase (** *p* < 0.01)), and Chi–Au 100 ppm (Day 3: ~1.3-fold increase (* *p* < 0.05), Day 5: ~1.22-fold increase, Day 7: ~2.5-fold increase (*** *p* < 0.001)); vWF: (control (1-fold increase at 3, 5, and 7 days), Chi (Day 3: ~1.3-fold increase (* *p* < 0.05), Day 5: ~1.78-fold increase (** *p* < 0.01), Day 7: ~1.47-fold increase (* *p* < 0.05)), Chi–Au 25 ppm (Day 3: ~1.37-fold increase (* *p* < 0.05), Day 5: ~1.47-fold increase (* *p* < 0.05), Day 7: ~2.24-fold increase (** *p* < 0.01)), Chi–Au 50 ppm (Day 3: ~1.29-fold increase (* *p* < 0.05), Day 5: ~1.49-fold increase (* *p* < 0.05), Day 7: ~1.5-fold increase (* *p* < 0.05)), and Chi–Au 100 ppm (Day 3: ~1.24-fold increase (* *p* < 0.05), Day 5: ~1.40-fold increase, Day 7: ~1.41-fold increase). The above evidence from real-time PCR analysis, FACS method, and immunofluorescence staining indicate that Chi combined with 50 ppm of Au nanoparticles could induce MSCs to undergo various differentiation routes, particularly for neural differentiation.

### 3.8. Subcutaneous Implantation in an Animal Model for Biocompatibility Measurements

Implanting biomaterials into tissues may cause chronic inflammation, thereby decreasing the efficiency of regeneration. Therefore, in our research, rats received subcutaneous implantations of various nanocomposite coatings to further investigate the biocompatibility of these compounds. Figure 7A depicts the capsule formation as disclosed by H&E staining. Figure 7D shows the quantified data of capsule thickness, which are as follows: control (1-fold increase), Chi ~ 0.84-fold increase (* *p* < 0.05), Chi–Au 25 ppm ~0.4-fold increase (** *p* < 0.01, # *p* < 0.05), Chi–Au 50 ppm ~0.38-fold increase (** *p* < 0.01, # *p* < 0.05), and Chi–Au 100 ppm ~0.58-fold increase, (** *p* < 0.01). Figure 7B,E display the collagen deposition evaluated by Masson’s trichrome staining, and the data are described as follows: control 1-fold increase, Chi ~0.85-fold increase (* *p* < 0.05), Chi–Au 25 ppm ~0.42-fold increase (** *p* < 0.01, # *p* < 0.05), Chi–Au 50 ppm ~0.39-fold increase (** *p* < 0.01, # *p* < 0.05), and Chi–Au 100 ppm ~0.56-fold increase (** *p* < 0.01, # *p* < 0.05). Figure 7C,F show the CD31 endothelialization revealed by AEC staining, and the results are also semi-quantified as follows: control 1-fold increase, Chi ~1.08-fold increase (* *p* < 0.05), Chi–Au 25 ppm ~ 1.15-fold increase (** *p* < 0.01, # *p* < 0.05), Chi–Au 50 ppm ~1.18-fold increase (** *p* < 0.01, # *p* < 0.05), and Chi–Au 100 ppm ~1.13-fold increase (* *p* < 0.05).

Furthermore, the M1 (CD86)/M2 (CD163) macrophage polarization (Figure 8A,B) and CD45 leukocyte filtration (Figure 8C) influenced by the Chi–Au nanocomposites were also examined. Figure 8D shows the quantification of CD86 expression intensity for M1 polarization as follows: control 1-fold increase, Chi ~0.82-fold increase (* *p* < 0.05), Chi–Au 25 ppm ~0.42-fold increase (** *p* < 0.01, # *p* < 0.05), Chi–Au 50 ppm ~0.4-fold increase (** *p* < 0.01, # *p* < 0.05), and Chi–Au 100 ppm ~0.66-fold increase (** *p* < 0.01, # *p* < 0.05). The quantification of the CD163 expression’s intensity (Figure 8E) for M2 polarization was as follows: control 1-fold increase, Chi ~1.12-fold increase (* *p* < 0.05), Chi–Au 25 ppm ~1.35-fold increase (** *p* < 0.01, # *p* < 0.05), Chi–Au 50 ppm ~1.4-fold increase (** *p* < 0.01, # *p* < 0.05), and Chi–Au 100 ppm ~1.28-fold increase (** *p* < 0.01). Additionally, the CD45 expression was quantified to determine leukocyte filtration as shown in Figure 8F. The results were evaluated as control 1-fold increase, Chi ~0.82-fold increase (* *p* < 0.05), Chi–Au 25 ppm ~0.72-fold increase (** *p* < 0.01, # *p* < 0.05), Chi–Au 50 ppm ~0.7-fold increase (** *p* < 0.01, # *p* < 0.05), and Chi–Au 100 ppm ~0.78-fold increase, * *p* < 0.05. The above histology and immunohistochemistry (IHC) results demonstrate that subcutaneous implantation of Chi-Au 50 ppm significantly decreased capsule thickness, reduced collagen deposition, inhibited the inflammatory response, and remarkably enhanced endothelialization.

A summary illustration is shown in Figure 9. In vitro assessments indicated that the combination of Chi with Au 50 ppm significantly strengthened the biological function and neural differentiation of MSCs. Moreover, in vivo subcutaneous implantation demonstrated that Chi–Au 50 ppm had better biocompatibility than the other tested nanomaterials. In conclusion, the above findings support the other results showing that Chi–Au 50 ppm has considerable potential as a biomaterial for neural tissue regeneration.

## 4. Discussion

The human nerve growth rate can reach approximately 2–5 mm per day [45], which is considered to be slow for nerve regeneration. Therefore, there is a need to develop a highly safe and effective drug to enhance nerve growth in clinical treatments. Nanotechnology has been applied in the development of appropriate biomaterials for regeneration treatments [46]. Artificial implants such as tubular silicone prostheses can be used for clinical nerve engineering. However, the use of non-biodegradable items within the body is a potential disadvantage in clinical treatments since they may require a second surgery for removal [47]. In recent decades, to improve the efficiency of therapeutic approaches in nerve tissue engineering, various natural and synthetic biomaterials containing bioactive molecules have been well investigated. Suitable nerve-repairing biomaterials must possess low cytotoxicity, exhibit good biocompatibility, induce as little inflammation as possible, and provide a favorable microenvironment for nerve growth [47]. Natural polymers such as chitosan and collagen have been used as nanofiber tubes for clinical applications. In a previous study, collagen-GAG-filled collagen tubes were assessed in vivo. The results demonstrated a continuous layer of myofibroblasts, and axonal regrowth was more effective [48]. Furthermore, chitosan was used to form nanofiber mesh tubes in another investigation. The findings showed the chitosan scaffold exhibited superior advantages such as enhanced permeability and facilitation of humoral permeation, thereby strengthening cell migration for nerve tissue repair [49].

Au nanoparticles have been shown to have potential in various medical fields, including tissue engineering and regenerative medicine [50]. Au also possesses unique physicochemical properties such as low cytotoxicity, and it can be synthesized easily [51]. Based on the FTIR results in our study (Figure 1D), the absorption peak at around 3600-3400 cm^−1^ represented an O–H stretching bond, which plays a vital role for the stability of Chi–Au nanoparticles in an aqueous solution [52]. Additionally, the amide I (1640 cm^−1^, C=O stretching) and amide II (1540 cm^−1^, N–H bending) bands [53] were found to be significantly changed after the combination of Chi and Au nanoparticles (Figure 1D). The covalent bonding of the amide I group represented the carboxylic group, which shifted due to the linkage of the Chi and Au nanoparticles [54]. Simultaneously, the shifted peak of N–H bending indicated the bonding of the Au to nitrogen atoms [54,55]. The changes from amide bands were due to the electrostatic interactions between the particle surface of the Au and amide groups [56]. The results verify that Au nanoparticles are coupled with chitosan in our research. Furthermore, Au nanoparticles can modify the surface topography of natural and synthetic polymers, conferring favorable properties at a low concentration owing to their extremely small size, i.e., at the nanoscale (5–50 nm) [57], leading to the improvement of surface physicochemical properties. The surface roughness would influence the cell aggregation due to the interaction between the nanocomposites and cells [58,59]. According to our AFM images (Appendix A), the RMS values of the Chi–Au 50 and 100 ppm nanocomposites were 0.67 and 0.38 nm, indicating a uniform surface topography for cell aggregation. In addition, the superior physicochemical properties of Au (~5 nm) can facilitate the electrical conductivity and strengthen the cell–cell interaction [60], which corresponded to the results of enhancing the MSCs’ proliferation and differentiation capacity in our study (Figure 2A,B).

The versatile applications of Au nanoparticles can create appropriate nanocomposite scaffolds to enhance cell viability and differentiation capacities [61]. A recent study demonstrated that a chitosan/κ-carrageenan hydrogel modified with Au nanoparticles strengthened the viability and attachment of MD-63 cells [62]. A major goal in the development of biomaterials is to enhance the differentiation capacity of stem cells using biocompatible Au-modified nanocomposites. For example, embryonic stem cells (ESCs) and mesenchymal stem cells (MSCs) possess superior self-renewal abilities and can differentiate into various cell types (neurons and endothelial cells) through the induction of cell signals [63]. It has been shown that Au nanoparticles could facilitate the differentiation of ESCs into dopaminergic (DA) neurons, and an association has been shown between the mTOR/p70S6K signaling pathway and differentiation of dopaminergic (DA) neurons [64]. Furthermore, FGF2-mediated and RA (retinoic acid)-mediated differentiated pathways were also found to be related to the neuronal differentiation of ESCs induced by Au-modified polymers [65]. Previous research has proven that the Chi–Au conduit can stimulate the expression of the brain-derived neurotrophic factor (BDNF) and glial cell line-derived neurotrophic factor (GDNF) in NSCs for nerve tissue regeneration [66]. However, NSCs are difficult to harvest. Additionally, they also need bioactive molecules to support proliferation [67]. Adipose-derived stem cells (ASCs) have also been applied for nerve regeneration [68]. The limitations of ASCs do exist, for example, the lack of self-renewal ability and risk of malignant transformation [69]. Therefore, MSCs are a promising cell resource for nerve repair.

Nestin is a type VI intermediate filament protein that is expressed in nerve cells for the radial growth of axon. Nestin is also expressed in the development of CNS [70]. Furthermore, nestin is found to express in NSPCs and astroglia cells [71]. GFAP is the glial fibrillary acidic protein, which is the marker of glial cells. GFAP was found to increase expression, while stem cells differentiate into glial cells [72]. The microtubules are the cytoskeleton composed of α and β-tubulin, which are associated with the cell migration of newly developed neurons [73]. MSCs are a cell resource with multiple differentiation capacities. Therefore, the investigation of the nestin, GFAP, and β-tubulin expressions in MSCs plays a vital role during neural differentiation when culturing with Chi–Au nanocomposites. Based on the results in our study, Chi–Au 50 ppm induced a higher mRNA level of neural differentiation markers (GFAP, β-tubulin, and nestin) in MSCs using real-time PCR analysis (Figure 5A) and FACS method (Figure 6A). Au nanoparticles were shown to inhibit ROS generation in MSCs [74], which is also consistent with our results (Figure 3E). The matrix metalloproteinase could cause the remodeling of ECM for cell migration. The higher expression level of MMP-9 proteins was verified to facilitate NSPCs’ differentiating into migratory cells for neural regeneration [75]. The results shown in Figure 4A,B indicate Chi–Au nanocomposites induced MMP-9 expression in MSCs, particularly at a concentration of 50 ppm Au. Moreover, based on the FACS result exhibited in Figure 2C, the expression of CD44 in MSCs was found to be significantly lower in the Chi–Au 50 ppm treatment. Previous research has elucidated that the expression of β-tubulin neuronal markers in bone-marrow MSCs increased, while the CD44 expression decreased [76]. Our results confirmed that Chi–Au 50 ppm could facilitate MSCs’ undergoing differentiation.

In our study, chitosan was fabricated using various concentrations of Au nanoparticles to investigate their potentials for neural regeneration in tissue engineering. Based on our findings, the Chi–Au nanocomposites demonstrated better induction of MSC colony formation, while the lower expression of a CD44 stem cell marker indicating Chi–Au could stimulate MSCs to differentiate. The biocompatibility assays indicated that Chi combined with 50 ppm of Au nanoparticles stimulated the lowest activation of an inflammatory response. The Chi–Au 50 ppm also induced the highest expression of neural differentiation markers (GFAP, β-tubulin, and nestin) in MSCs. Additionally, the results from the in vivo assessments supported the other results, showing that Chi–Au 50 ppm is a promising biomaterial for clinical treatments. Therefore, we recommend the combination of Chi–Au 50 ppm nanocomposites and MSCs as a potential therapeutic approach for nerve regeneration.

## 5. Conclusions

In our research, Chi was fabricated with various concentrations of Au nanoparticles (25, 50, and 100 ppm). Based on the results of in vitro assessments, Chi–Au 50 ppm demonstrated better biocompatibility and significantly strengthened MSC colony formation. The Chi–Au 50 ppm facilitated the matrix metalloproteinase 9 (MMP-9) and vinculin expression. Additionally, Chi combined with 50 ppm of Au significantly stimulated neural markers (GFAP, β-tubulin, and nestin) to express, thereby promoting neural differentiation in MSCs. The results of subcutaneous implantation of Chi–Au 50 ppm in a rat model showed decreased capsule formation and inhibition of inflammatory response. The endothelialization capabilities were also facilitated by Chi–Au 50 ppm. Therefore, taken together, our findings provide evidence that Chi–Au 50 ppm is a fascinating candidate for neural tissue engineering.

## Figures and Tables

**Figure 1 cells-11-01861-f001:**
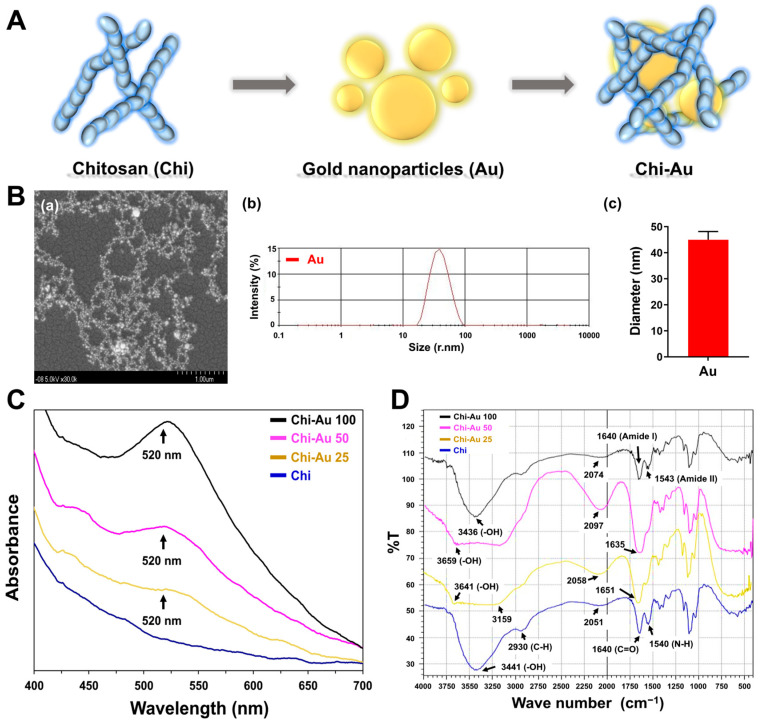
Characterization of the as-prepared materials. (**A**) Brief concept of preparing Chi–Au nanocomposites. (**B**) The (a) SEM image and (b) DLS assay for identification of the Au nanoparticles. Scale bar was set as 1 μm. (c) The diameter of the Au nanoparticles was 45 ± 3.2 nm. (**C**) UV-Vis spectrum for Chi fabricated with various concentrations of Au nanoparticles (25, 50, and 100 ppm). The peak at 520 nm indicated as Au. (**D**) The FTIR spectrum for Chi–Au (25, 50, and 100 ppm) nanocomposites ranged from 400 to 4000 cm^−1^ wavenumber. The results are represented as three independent experiments.

**Figure 2 cells-11-01861-f002:**
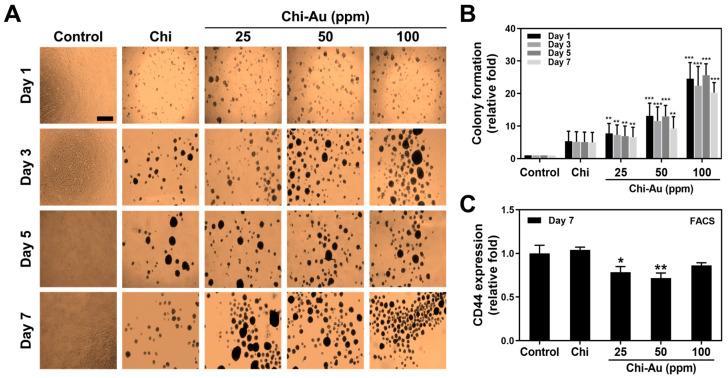
Colony formation and CD44 expression of MSCs cultured on Chi–Au nanocomposites. (**A**) Colony formation ability and (**B**) colony formation number of MSCs were observed at 1, 3, 5, and 7 days. Furthermore, (**C**) the expression of the CD44 marker at Day 7 was semi-quantified by the fluorescence-activated cell sorting (FACS) method. The results demonstrated that Chi combined with Au could induce the proliferation capacities of MSCs, particularly at the concentrations of 50 and 100 ppm of Au nanoparticles. Additionally, the CD44 expression was explored to be significantly lower in the Chi–Au 25 and 50 ppm groups. The scale bar to observe colony formation was 100 μm. Data are exhibited as the mean ± SD (*n* = 3). * *p* < 0.05, ** *p* < 0.01, *** *p* < 0.001: compared with the control (TCPS).

**Figure 3 cells-11-01861-f003:**
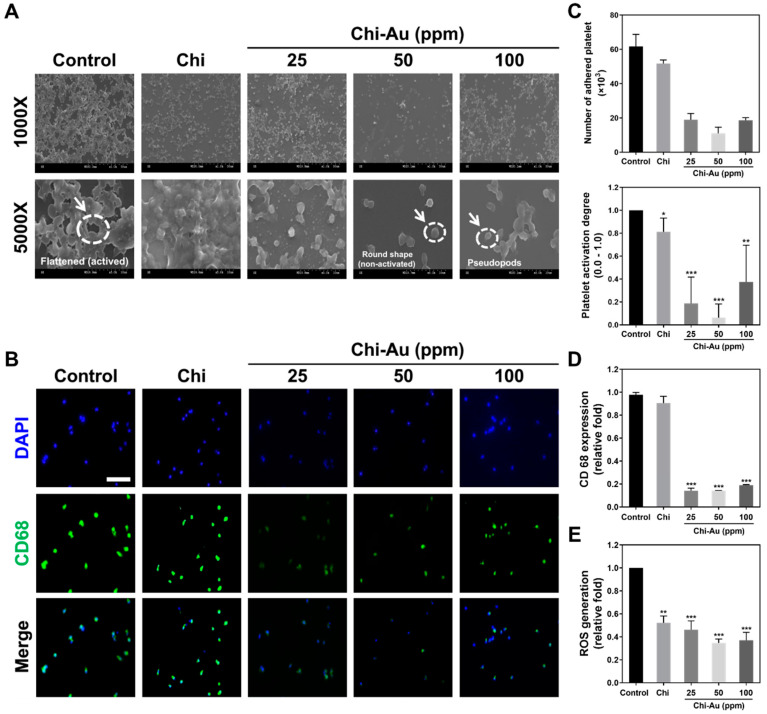
Biocompatibility assessments of the Chi–Au nanocomposites. (**A**) The platelet activation culturing on various nanocomposites was observed by SEM after 24 hours’ incubation. The flattened platelet cells constituted the activated form (white arrow in the Control), while the non-activated platelet cells were observed to have a round shape (white arrow in the Chi–Au 50 ppm). The white arrow in Chi–Au 100 ppm indicated pseudopod morphology of platelet cell. (**B**) The CD68 macrophage marker expressed in monocytes was detected through IF staining after 96 h. The monocytes were immunostained with primary anti-CD68 antibodies and conjugated with secondary FITC antibodies (green fluorescence). DAPI solution was used to detect nuclei (blue fluorescence). Scale bars measure 20 μm. (**C**) The number of adhered platelet cells and degree of activation were quantified. Based on the results, Chi combined with 50 ppm of Au nanoparticles induced the lowest amount of adherence and degree of activation of the platelet cells. (**D**) The expression of CD68 in the monocytes was semi-quantified based on fluorescence density. The expression of CD68 was significantly decreased in the Chi–Au groups. (**E**) The intracellular ROS generation of MSCs seeded on various nanocomposites was targeted by DCFH-dA probe and semi-quantified through flow cytometry. Data are represented by one of three independent experiments. * *p* < 0.05, ** *p* < 0.01, *** *p* < 0.001: compared with the control (TCPS).

**Figure 4 cells-11-01861-f004:**
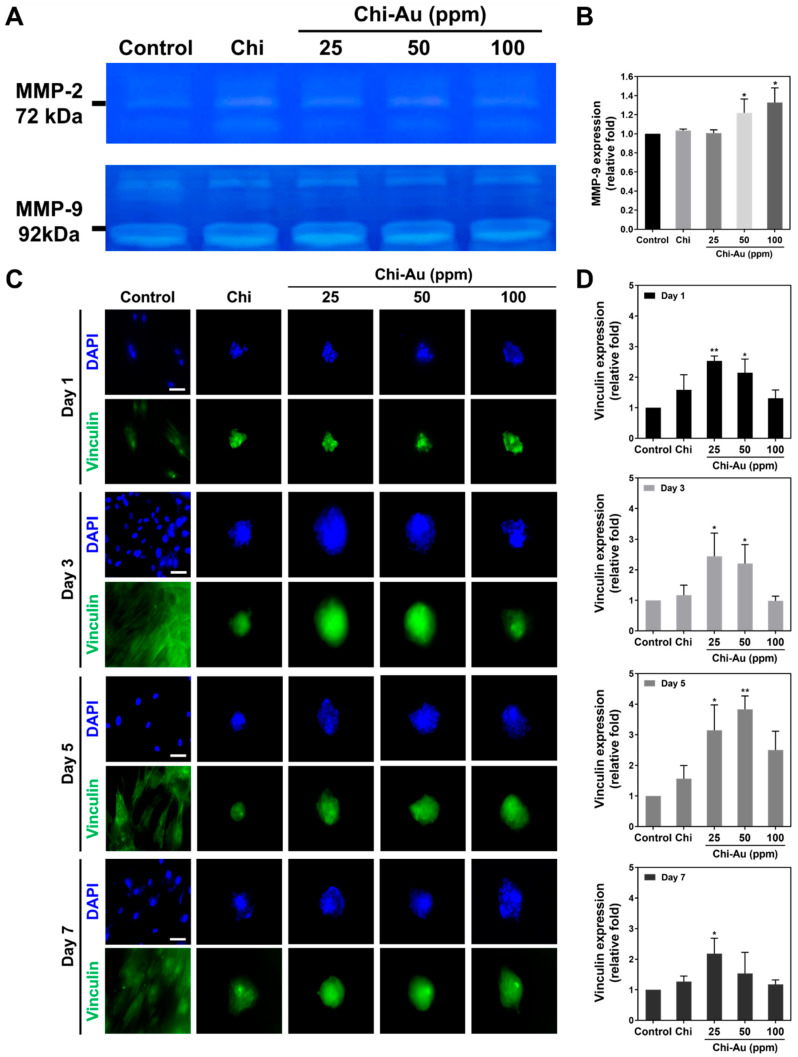
Expression of MMP-2/9 protein and vinculin in MSCs incubated with Chi–Au nanocomposites. (**A**) The MMP-2/9 protein expressions at 48 h are shown. (**B**) The MMP-9 protein expression was semi-quantified. The MMP-9 protein expression was significantly increased in the Chi–Au 50 and 100 ppm groups. (**C**) The images of vinculin expression in MSCs were demonstrated by immunofluorescence staining at 1, 3, 5, and 7 days. Scale bars measure 20 μm. (**D**) The expression of vinculin in MSCs at various time points was semi-quantified according to expression intensity. The vinculin expression was significantly higher when treated with Chi–Au 50 ppm at 5 days. Data are displayed as mean ± SD (*n* = 3). * *p* < 0.05, ** *p* < 0.01: compared with the control (TCPS).

**Figure 5 cells-11-01861-f005:**
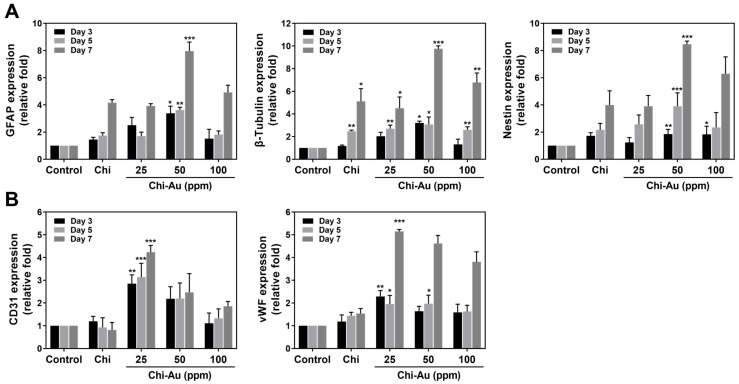
The mRNA expression of various differentiation markers in MSCs cultured on Chi–Au nanocomposites was investigated using real-time PCR at 3, 5, and 7 days. (**A**) The expression of the neural differentiation markers GFAP, β-Tubulin and nestin were semi-quantified. The results indicated that Chi–Au 50 ppm significantly induced neural differentiation. (**B**) The expression of the endothelialization markers CD31 and vWF were semi-quantified. CD31 and vWF mRNA were significantly expressed when treated with Chi–Au 25 ppm. The results are demonstrated as mean ± SD (*n* = 3). * *p* < 0.05, ** *p* < 0.01, *** *p* < 0.001: compared with the control (TCPS). Tabulation Data of mRNA Expression Induced by Chi–Au Nanocomposites.

**Figure 6 cells-11-01861-f006:**
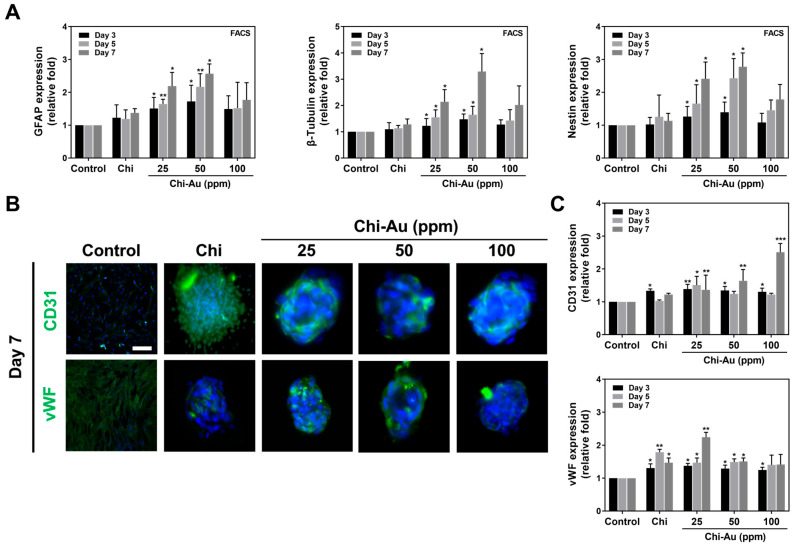
Differentiation capacities of MSCs induced by Chi–Au nanocomposites were measured by fluorescence-activated cell sorting (FACS) and immunofluorescence (IF) staining at 3, 5, and 7 days. The cells were first stained with primary antibodies (GFAP, β-Tubulin, nestin, CD31, and vWF) and conjugated with FITC immunoglobulin second antibodies (green fluorescence). The cell nuclei were detected by DAPI (blue fluorescence). (**A**) Quantification of the neural differentiation protein expressions based on the FACS method. The results indicate that Chi–Au 50 ppm nanocomposites significantly induced the expression of GFAP, β-Tubulin, and nestin. (**B**) Images of endothelialization markers’ (CD31 and vWF) expression at day 7 are shown. (**C**) The expression of endothelialization markers was semi-quantified for 3, 5, and 7 days. Based on the quantified data, both CD31 and vWF expression were found to be higher in the Chi–Au 25 ppm and 50 ppm groups. Scale bar measures 100 μm. Results are indicated as mean ± SD (*n* = 3). * *p* < 0.05, ** *p* < 0.01, *** *p* < 0.001: compared with the control (TCPS).

**Figure 7 cells-11-01861-f007:**
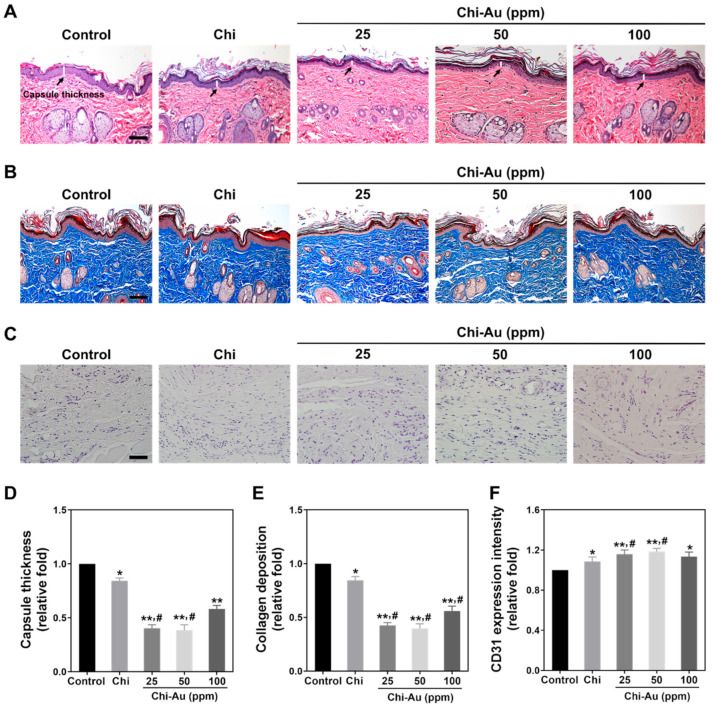
Histology assessments of biocompatibility in an animal model after subcutaneous implantation for four weeks. (**A**) Capsule thickness was observed by H&E staining. The black arrows show the capsule thickness induced by various Chi–Au nanocomposites. (**B**) Collagen deposition was examined by Masson’s trichrome staining (blue). (**C**) CD31 endothelial marker expression was detected by AEC staining. (**D**,**E**) The capsule thickness and collagen deposition in each group were semi-quantified. The results demonstrate that Chi–Au 50 ppm induced the lowest capsule formation and collagen deposition, indicating that Chi–Au 50 ppm would not significantly stimulate a foreign body response. (**F**) CD31 expression was semi-quantified. The Chi–Au 25 ppm and 50 ppm groups remarkably induced endothelialization. Scale bar was set as 100 μm. Results are exhibited as mean ± SD (*n* = 3). * *p* < 0.05, ** *p* < 0.01: compared with the control (TCPS); # *p* < 0.05: compared with the Chi group.

**Figure 8 cells-11-01861-f008:**
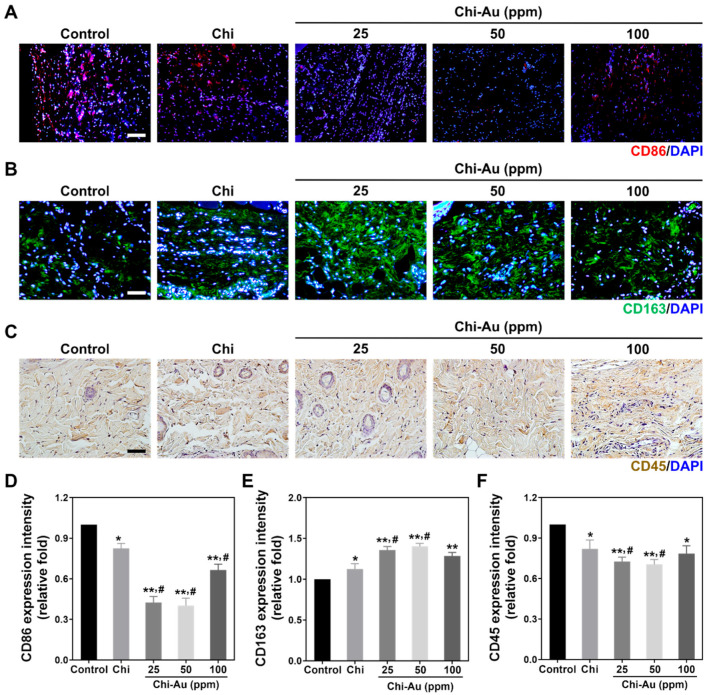
Immunohistochemical (IHC) staining for the investigation of anti-inflammatory capacities after subcutaneous implantation for 4 weeks. (**A**,**B**) The expressions of CD86 (M1 polarization, red color) and CD163 (M2 polarization, green color) are shown. (**C**) The expression of CD45, which indicates leukocyte filtration, was observed using DAB staining. (**D**,**E**) The expressions of CD86 and CD163 were semi-quantified. Chi combined with 50 ppm of Au stimulated the lowest CD86 expression. In contrast, the expression of CD163 was remarkably greater with Chi–Au 50 ppm treatment. (**F**) The CD45 expression was quantified. Based on the result, Chi–Au 50 ppm induced the lowest CD45 expression. The results elucidate that Chi with 50 ppm of Au exhibited better biocompatibility when compared with the other tested nanocomposites. Scale bar was 100 μm. Results are presented as mean ± SD (*n* = 3). * *p* < 0.05, ** *p* < 0.01: compared with the control (TCPS); # *p* < 0.05: compared with the Chi group.

**Figure 9 cells-11-01861-f009:**
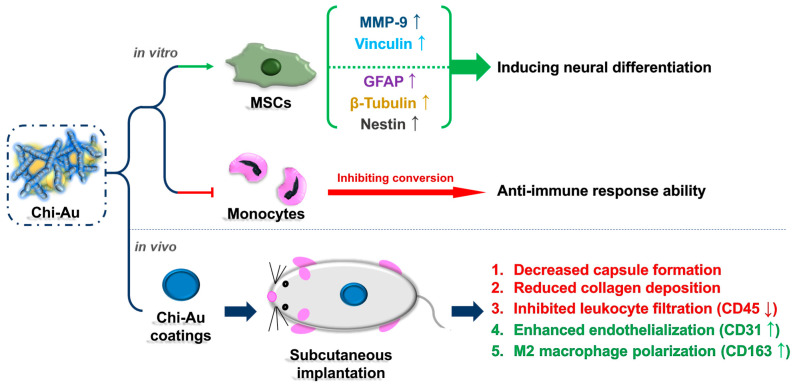
Summary illustrations for Chi–Au nanocomposites in in vitro and in vivo assessments. The summary shows that Chi combined with 50 ppm of Au nanoparticles can significantly induce neural differentiation. Furthermore, the animal model also indicated Chi–Au 50 ppm was a highly safe nanocomposite, indicating its potential for use in tissue engineering. The up and down arrows indicated the increased or decreased expression of protein and mRNA.

**Table 1 cells-11-01861-t001:** GFAP mRNA expression at 3, 5, and 7 days.

GFAP mRNA Expression (Relative Fold)	Day 3	Day 5	Day 7
Control	1	1	1
Chi	~1.44	~1.74	~4.16
Chi–Au 25 ppm	~2.5	~1.71	~3.92
Chi–Au 50 ppm	~3.37 *	~3.61 **	~7.96 ***
Chi–Au 100 ppm	~1.51	~1.81	~4.92

Note: * *p* < 0.05, ** *p* < 0.01, *** *p* < 0.001: compared with the control (TCPS).

**Table 2 cells-11-01861-t002:** β-Tubulin mRNA expression at 3, 5, and 7 days.

β-Tubulin mRNA Expression (Relative Fold)	Day 3	Day 5	Day 7
Control	1	1	1
Chi	~1.17	~2.48 **	~5.1 *
Chi–Au 25 ppm	~2.02	~2.7 **	~4.49 *
Chi–Au 50 ppm	~3.2 *	~3.07 *	~9.76 ***
Chi–Au 100 ppm	~1.3	~2.59 **	~6.77 **

Note: * *p* < 0.05, ** *p* < 0.01, *** *p* < 0.001: compared with the control (TCPS).

**Table 3 cells-11-01861-t003:** Nestin mRNA expression at 3, 5, and 7 days.

Nestin mRNA Expression (Relative Fold)	Day 3	Day 5	Day 7
Control	1	1	1
Chi	~1.72	~2.16	~3.98
Chi–Au 25 ppm	~1.23	~2.56	~3.89
Chi–Au 50 ppm	~1.84 **	~3.89 ***	~8.46 ***
Chi–Au 100 ppm	~1.82 *	~2.34	~6.29

Note: * *p* < 0.05, ** *p* < 0.01, *** *p* < 0.001: compared with the control (TCPS).

**Table 4 cells-11-01861-t004:** CD31 mRNA expression at 3, 5, and 7 days.

CD31 mRNA Expression (Relative Fold)	Day 3	Day 5	Day 7
Control	1	1	1
Chi	~1.19	~0.93	~0.81
Chi–Au 25 ppm	~2.84 **	~3.13 ***	~4.23 ***
Chi–Au 50 ppm	~2.18	~2.19	~2.47
Chi–Au 100 ppm	~1.11	~1.33	~1.86

Note: ** *p* < 0.01, *** *p* < 0.001: compared with the control (TCPS).

**Table 5 cells-11-01861-t005:** vWF mRNA expression at 3, 5, and 7 days.

vWF mRNA Expression (Relative Fold)	Day 3	Day 5	Day 7
Control	1	1	1
Chi	~1.18	~1.44	~1.54
Chi–Au 25 ppm	~2.28 **	~1.95 *	~5.14 ***
Chi–Au 50 ppm	~1.63	~1.97 *	~4.61
Chi–Au 100 ppm	~1.58	~1.62	~3.8

Note: * *p* < 0.05, ** *p* < 0.01, *** *p* < 0.001: compared with the control (TCPS).

## Data Availability

Data are contained within the article.

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
