# Peer review of "Neural Differentiation Potential of Mesenchymal Stem Cells Enhanced by Biocompatible Chitosan-Gold Nanocomposites"

_cells, 2022, doi:10.3390/cells11121861_

Round 1

Reviewer 1 Report

Dear Authors

in this paper, you use Chitosan coupled with different concentrations (25, 50, and 100 ppm) of gold (Au) nanoparticles for use in in vitro and in vivo experiments. In summary, Chi polymer combined with 50 ppm of Au nanoparticles was proven to enhance the neural differentiation of MSCs and showed potential as a biosafe nanomaterial for neural tissue engineering. I find the paper interesting but in my opinion, the author needs to clarify several aspects. I believe that the paper could be accepter only after major revision: 

  •  the authors conclude that the compound (Chi-au 50) is able to maintain stemness and at the same time is able to induce neuronal differentiation of mesenchymal stem cells. These data are somewhat in contradiction.
    I ask the authors to perform a flow cytometric analysis to verify the expression of CD44 and CD105 under control conditions and after stimulation with the different combinations of Chi-au. This experiment can clarify whether the compound is truly able to maintain the stemness properties of mesenchymal stem cells.
  • the behavior of nestin is not clear. As the authors know this molecule is a marker of neuronal stem cells. In Figure 7 C the authors reported that increases at 7 days (Chi-au 50) together with GFAP. it is very strange, when cells differentiate nestin decreases, and neuronal markers increase (GFAP, B3-tubulin, NFH, etc.). Furthermore, B3-tubulin does not increase significantly in the samples (Chi-au 50).
    I ask the authors to perform a flow cytometric analysis to verify the expression of B3-tubulin,  GFAP, and nestin under control conditions and after stimulation with different combinations of Chi-au and with the addition of a long time (for example 12-14 days).
  • in the abstract, the authors report that Chi-Au 50 ppm was also discovered to significantly induce the expression of GFAP, β-tubulin, and nestin protein in MSCs for neural differentiation and never mention the role of Chi-au in osteogenic differentiation. What sense does it make to insert the data in figure 8 regarding osteogenic differentiation? Sometimes less is better.

minor revision:

  • for a better reading of the paper, I suggest deleting figure 2 and inserting it in the supplementary material.
  • I advise once the flow cytometric analyzes have been carried out, replace them with the respective figures 3B and 7AC
  • I recommend adding in the introduction (line 104) a description of the different types of mesenchymal stem cells (dental pulp, adipose tissue, etc.) with relative references

Author Response

Reviewer 1:

in this paper, you use Chitosan coupled with different concentrations (25, 50, and 100 ppm) of gold (Au) nanoparticles for use in in vitro and in vivo experiments. In summary, Chi polymer combined with 50 ppm of Au nanoparticles was proven to enhance the neural differentiation of MSCs and showed potential as a biosafe nanomaterial for neural tissue engineering. I find the paper interesting but in my opinion, the author needs to clarify several aspects. I believe that the paper could be accepter only after major revision: 

  1. the authors conclude that the compound (Chi-au 50) is able to maintain stemness and at the same time is able to induce neuronal differentiation of mesenchymal stem cells. These data are somewhat in contradiction.
    I ask the authors to perform a flow cytometric analysis to verify the expression of CD44 and CD105 under control conditions and after stimulation with the different combinations of Chi-au. This experiment can clarify whether the compound is truly able to maintain the stemness properties of mesenchymal stem cells.
    Answer:
    Thanks for the comment from the reviewer. We have performed the FACS data for CD44 expression in the new Figure 2C.

Some descriptions have been revised as followed:
(1) The new description is included in the “Results” section. (Page 8, Line 354-365)
“CD44 expression in MSCs colonies at 7 days was further investigated by fluorescence activated cell sorting (FACS) method. The results based on CD44 expressed intensity are depicted in Figure 2C. Expression of CD44 in the control group showed a 1-fold increase at 7 days. In the pure Chi group, the result was not significantly different compared to the control [Day 7: ~ 1.04-fold increase]. Chi-Au 25 ppm treatment induced lower CD44 expression in MSCs [Day 7: ~ 0.78-fold, (*p < 0.05)]. In the Chi-Au 50 ppm group, the expression of CD44 was also significantly lower than in the control [Day 7: ~ 0.71-fold (**p < 0.01)]. Furthermore, CD44 expression in the Chi-Au 100 ppm group was no significant difference compared with the control [Day 7: ~ 0.86-fold]. Based on the above evidence, Chi-Au nanocomposites could significantly enhance MSCs proliferation. Furthermore, the lower expression of CD44 was significantly lower in Chi-Au treatment, indicating that Chi-Au nanocomposites could induce the MSCs undergo differentiation.”
(2) The new description is included in the “Figure caption” section. (Page 9, Line 367-373)

Figure 2. Colony formation and CD44 expression of MSCs cultured on Chi-Au nanocomposites. (A) Colony formation ability and (B) colony formation number of MSCs were observed at 1, 3, 5, and 7 days. Furthermore, (C) the expression of CD44 marker at Day 7 was semi-quantified by fluorescence activated cell sorting (FACS) method. The results demonstrated Chi combined with Au could induce proliferation capacities of MSCs, particularly at the concentration 50 and 100 ppm of Au nanoparticles. Additionally, the CD44 expression was explored to be significantly lower in Chi-Au 25 and 50 ppm group….”
(3) The new description is included in the “Discussion” section..
“Moreover, based on the FACS result exhibited in Figure 2C, the expression of CD44 in MSCs was found to significantly lower in the Chi-Au 50 ppm treatment. Previous research has elucidated that the expression of β-tubulin neuronal markers in bone-marrow MSCs increased while the CD44 expression decreased [75]. Our results confirmed that Chi-Au 50 ppm could facilitate MSCs undergo differentiation.” (Page 20, Line 687-691)
“Chi-Au nanocomposites demonstrated better induction of MSC colony formation, while the lower expression of CD44 stem cell marker indicating Chi-Au could stimulate MSCs to differentiate.” (Page 21, Line 694-696)
Reference:
75. Tao, H.; Rao, R.; Ma, D.D. Cytokine‐induced stable neuronal differentiation of human bone marrow mesenchymal stem cells in a serum/feeder cell‐free condition. Development, growth & differentiation 2005, 47, 423-433.
(4) The new description is included in the “Abstract” section. (Page 1, Line 25-29)
“Furthermore, Chi-Au 50 ppm could facilitate colony formation, and strengthen matrix metalloproteinase (MMP) activation in mesenchymal stem cells (MSCs). The lower expression of CD44 in Chi-Au 50 ppm treatment demonstrated that the nanocomposites could enhance the MSCs undergo differentiation.”

(5) The new description is included in the “Materials and Methods” section. (Page 6, Line 271-274)
“Additionally, CD44, GFAP, Nestin and β-Tubulin fluorescein-positive cells were identified by fluorescence activated cell sorting (FACS) Calibur flow cytometer (BD Biosciences, USA). The data were analyzed by version 7.6.1 Flow J software. All experiments were displayed in triplicate.”

(6) The title of present study has also been revised.
“Neural Differentiation Potential of Mesenchymal Stem Cells Enhanced by Biocompatible Chitosan-Gold Nanocomposites” (Page 1)

  1. the behavior of nestin is not clear. As the authors know this molecule is a marker of neuronal stem cells. In Figure 7 C the authors reported that increases at 7 days (Chi-au 50) together with GFAP. it is very strange, when cells differentiate nestin decreases, and neuronal markers increase (GFAP, B3-tubulin, NFH, etc.). Furthermore, B3-tubulin does not increase significantly in the samples (Chi-au 50).
    I ask the authors to perform a flow cytometric analysis to verify the expression of B3-tubulin,  GFAP, and nestin under control conditions and after stimulation with different combinations of Chi-au and with the addition of a long time (for example 12-14 days).
    Answer:
    Thanks for the comment from the reviewer.
    (1) We have included new description in the “Discussion” section (Page 20, Line 676-684)
    “Nestin is type VI intermediate filament protein which expressed in nerve cells for the radial growth of axon. Nestin is also expressed in the development of CNS [69]. Furthermore, nestin is found to express in NSPCs and astroglia cells [70]. GFAP is the glial fibrillary acidic protein, which is the marker of glial cells. GFAP was found to increase expression while stem cells differentiate into glial cells [71]. The microtubules are the cytoskeleton composed of α and β-tubulin, which is associated with cell migration of newly developed neurons [72]. MSCs is the cell resource with multiple differentiation capacities. Therefore, the investigation of nestin, GFAP and β-tubulin expression in MSCs plays vital role during neural differentiation when culturing with Chi-Au nanocomposites.”
    Reference:
    69. Lendahl, U.; Zimmerman, L. B.; and McKay, R. D. CNS stem cells express a new class of intermediate filament protein. Cell199060, 585-595.
    70. Filippov, V.; Kronenberg, G.; Pivneva, T.; Reuter, K.; Steiner, B.; Wang, L.-P.; Yamaguchi, M.; Kettenmann, H.; Kempermann, G. Subpopulation of nestin-expressing progenitor cells in the adult murine hippocampus shows electrophysiological and morphological characteristics of astrocytes. Molecular and Cellular Neuroscience200323, 373-382.
    71. Chen, H.; Wu, H.; Yin, H.; Wang, J.; Dong, H.; Chen, Q.; Li, Y. Effect of photobiomodulation on neural differentiation of human umbilical cord mesenchymal stem cells. Lasers in medical science 2019, 34, 667-675.
    72. Sheikh, A.M.; Yano, S.; Tabassum, S.; Omura, K.; Araki, A.; Mitaki, S.; Ito, Y.; Huang, S.; Nagai, A. Alteration of neural stem cell functions in ataxia and male sterility mice: A possible role of β-tubulin glutamylation in neurodegeneration. Cells 2021, 10, 155
    (2) We have included the new FACS data for b-tubulin, GFAP, and nestin expression in the “Results” section.

Some descriptions were revised as followed:
“Additionally, the expression of neural and endothelial differentiation proteins in MSCs was also examined by fluorescence activated cell sorting (FACS) method (Figure 6A) and immunofluorescence staining (Figure 6B). The FACS quantification of neural differentiation protein expression was displayed in Figure 6A. Chi-Au 50 ppm nanocomposites were found to significantly enhance the expression of neural-related proteins in MSCs. The results are as follows: GFAP: [Control 1-fold increase at 3, 5, and 7 days), Chi [Day 3: ~ 1.22-fold increase, Day 5: ~ 1.19-fold increase, Day 7: ~ 1.37-fold increase], Chi-Au 25 ppm [Day 3: ~ 1.50-fold increase (*p < 0.05), Day 5: ~ 1.64-fold increase (**p < 0.01), Day 7: ~ 2.18 fold (*p < 0.05)], Chi-Au 50 ppm [Day 3: ~ 1.72-fold increase (*p < 0.05), Day 5: ~ 2.16-fold increase (**p < 0.01), Day 7: ~ 2.56-fold increase (*p < 0.05)], Chi-Au 100 ppm [Day 3: ~ 1.49-fold increase, Day 5: ~ 1.52-fold increase, Day 7: ~ 1.76-fold increase]; β-Tubulin: [Control 1-fold increase at 3, 5, and 7 days), Chi [Day 3: ~ 1.09-fold increase, Day 5: ~ 1.13-fold increase, Day 7: ~ 1.28-fold increase], Chi-Au 25 ppm [Day 3: ~ 1.22-fold increase (*p < 0.05), Day 5: ~ 1.55-fold increase (*p < 0.05), Day 7: ~ 2.13-fold increase (*p < 0.05)], Chi-Au 50 ppm [Day 3: ~ 1.47-fold increase (*p < 0.05), Day 5: ~ 1.64 -fold increase (*p < 0.05), Day 7: ~ 3.28-fold increase (*p < 0.05)], Chi-Au 100 ppm [Day 3: ~ 1.27-fold increase, Day 5: ~ 1.42-fold increase, Day 7: ~ 2.01-fold increase]; nestin: [Control (1-fold increase at 3, 5, and 7 days), Chi [Day 3: ~ 1.02-fold increase, Day 5: ~ 1.25-fold increase, Day 7: ~ 1.13-fold increase], Chi-Au 25 ppm [Day 3: ~ 1.26-fold increase (*p < 0.05), Day 5: ~ 1.65-fold increase (*p < 0.05), Day 7: ~ 2.41-fold increase (*p < 0.05)], Chi-Au 50 ppm [Day 3: ~ 1.39-fold increase (*p < 0.05), Day 5: ~ 2.42-fold increase (*p < 0.05), Day 7: ~ 2.77-fold increase (*p < 0.05)], Chi-Au 100 ppm [Day 3: ~ 1.08-fold increase, Day 5: ~ 1.45-fold increase, Day 7: ~ 1.78-fold increase] (Figure 6A).” (Page 14-15, Line 488-510)
“The images of endothelial differentiation protein expression at 7 days are demonstrated in Figure 6B. The images captured at 3 and 5 days are shown in Figure S5A. The semi-quantitative results of endothelial differentiation proteins expression based on fluorescence intensity are depicted in Figure 6C.” (Page 15, Line 511-514)
“The above evidence from real-time PCR analysis, FACS method and immunofluorescence staining indicate Chi combined with 50 ppm of Au nanoparticles could induce MSCs to undergo various differentiation routes, particularly for neural differentiation.” (Page 15, Line 527-530)
(4) We have included new description in the “Figure caption” section (Page 15-16, Line 531-541)
Figure 6. Differentiation capacities of MSCs induced by Chi-Au nanocomposites were measured by by fluorescence activated cell sorting (FACS) and immunofluorescence staining at 3, 5, and 7 days. The cells were first stained with primary antibodies (GFAP, β-Tubulin, nestin, CD31, and vWF) and conjugated with FITC immunoglobulin second antibodies (green fluorescence). Cell nuclei was detected by DAPI (blue fluorescence). (A) Quantification of neural differentiation protein expression based on FACS method. The results indicated Chi-Au 50 ppm nanocomposites significantly induced the expression of GFAP, β-Tubulin and nestin. (B) Images of endothelialization markers (CD31 and vWF) expression are shown. (C) The expression of endothelialization markers was semi-quantified for 3, 5, and 7 days. Based on the quantified data, both CD31 and vWF expression was found to be higher in the Chi-Au 25 ppm and 50 ppm group. Scale bar measures 100 μm. Results are indicated as mean ± SD (n = 3). *p < 0.05, **p < 0.01, ***p < 0.001: compared to the Control (TCPS).”
(5) We have included new description in the “Discussion” section. (Page 20, Line 680-682)
“…Chi-Au 50 ppm induced higher mRNA level of neural differentiation markers (GFAP, β-tubulin, and nestin) in MSCs using real-time PCR analysis (Figure 5A), and FACS method (Figure 6A) as well….”

  1. in the abstract, the authors report that Chi-Au 50 ppm was also discovered to significantly induce the expression of GFAP, β-tubulin, and nestin protein in MSCs for neural differentiation and never mention the role of Chi-au in osteogenic differentiation. What sense does it make to insert the data in figure 8 regarding osteogenic differentiation? Sometimes less is better.
    Answer:
    Thanks for the suggestion from the reviewer. We have removed the description associated with osteogenic differentiation in Abstract, Materials & Method (ARS staining), Results (section 3.6), and Figure 5C (Runx-2 mRNA expression by real-time PCR). Furthermore, the data related to osteogenic differentiation (ARS staining) is also deleted.

minor revision:

  1. for a better reading of the paper, I suggest deleting figure 2 and inserting it in the supplementary material.

Answer:

Thanks for the valuable suggestion from the reviewer. We have moved the original Figure 2 into new Figure S1.

  1. I advise once the flow cytometric analyzes have been carried out, replace them with the respective figures 3B and 7AC

Answer:

Thanks for the comment from the reviewer. We have replaced the FACS data for CD44 and neural differentiation markers differentiation in new Figure 2C and Figure 6A. The original immunostaining data for neural marker expression in Figure S5 has also been removed.

  1. I recommend adding in the introduction (line 104) a description of the different types of mesenchymal stem cells (dental pulp, adipose tissue, etc.) with relative references.

Answer:

Thanks for the valuable suggestion from the Reviewer. We have added a new description of MSCs in the “Introduction” section. The relative references are also included. (Page 3, Line 100-103)

“MSCs are abundant in the human body with the majority present in bone marrow, which can differentiate into various cell types with multifunctionality [33]. Furthermore, MSCs can also be obtained from adipose tissue [34], muscle [35], dental pulp [36] and umbilical blood [37].”

References:

  1. Samsonraj, R.M.; Raghunath, M.; Nurcombe, V.; Hui, J.H.; van Wijnen, A.J.; Cool, S.M. Concise review: multifaceted characterization of human mesenchymal stem cells for use in regenerative medicine. Stem cells translational medicine 2017, 6, 2173-2185.
  2. Aust, L.; Devlin, B.; Foster, S.; Halvorsen, Y.; Hicok, K.d.; Du Laney, T.; Sen, A.; Willingmyre, G.; Gimble, J. Yield of human adipose-derived adult stem cells from liposuction aspirates. Cytotherapy 2004, 6, 7-14.
  3. Jackson, W.M.; Nesti, L.J.; Tuan, R.S. Potential therapeutic applications of muscle-derived mesenchymal stem and progenitor cells. Expert opinion on biological therapy 2010, 10, 505-517.
  4. Shi, S.; Gronthos, S. Perivascular niche of postnatal mesenchymal stem cells in human bone marrow and dental pulp. Journal of bone and mineral research 2003, 18, 696-704.
  5. Corrao, S.; La Rocca, G.; Lo Iacono, M.; Corsello, T.; Farina, F.; Anzalone, R. Umbilical cord revisited: from Wharton’s jelly myofibroblasts to mesenchymal stem cells. 2013.

Reviewer 2 Report

Dear editor,

The manuscript (cells-1742602) reports synthesis of Chitosan-Au nanoparticles that can induce neural differentiation of MSCs. The authors have thoroughly investigated various biological properties of the hybrid system, particularly related to neural regeneration. The manuscript is organized and reads well. However, I have several major comments that need to be addressed to make the work re-considerable for publication:

1- Au nanoparticles are well-known for their neural regeneration capacity either alone or as coupled with chitosan. There are several similar or relevant studies in the literature such as: https://doi.org/10.1016/j.surneu.2008.01.057, https://journals.lww.com/jcma/Fulltext/2021/11000/Fabrication_of_hyaluronic_acid_gold_nanoparticles.6.aspx, https://doi.org/10.1016/j.bbrc.2019.03.189

Thus, at the first step, the authors should justify the novelty and significance of their study compared to the literature.

2- Too long introduction with unnecessary explanation about stem cells.

3- Page 3, line 124, why osteogenesis of MSCs should matter in this study dealing with neural tissue regeneration?

4- Page 4, line 180; how much was the electron accelerating voltage? any sputter coating?

5- Page 7, line 329-334; Au chelation should lead to disappearance of characteristic peaks of chitosan. The observed shifts can be mainly associated to H-bonding. Why it is so? describe the mechanism? 

6- line 339; why the reported values are so detailed? They should be rounded up/down.

7- The larger colonization of MSCs on the nanocomposite with higher density of Au nanoparticles could be related to enhanced electrical conductivity of the system that allows cell-cell communication. Can the authors comment on this aspect of their system?

8- Page 11, line 466; increased level of MMP-9 could be indicative of intensive inflammatory response, as well. Correct?

9- Page 14, line 507; the presentation of the results in the text format is confusing. A table or visual element that can summarize and present the data would be more appealing. 

10- The optimized biological performance of Chi-Au 50 ppm over that with higher ppm of Au nanoparticles could be attributed to which factors?

11- Figure 1Ba, SEM image is in poor quality, scale bar is hardly read and the dots are not clear what they represent?

12- Discussion; the statements about peripheral nerve injuries are unnecessary.  In general, discussion part does not analyze and discuss the achieved results rather repeats some general information about importance of nerve regeneration using nanomaterials, particularly Au nanoparticles, etc. The paper mainly reports the results and not scientifically discusses the reason behind the observed behaviours and the findings.

Author Response

Reviewer 2

Dear editor,

The manuscript (cells-1742602) reports synthesis of Chitosan-Au nanoparticles that can induce neural differentiation of MSCs. The authors have thoroughly investigated various biological properties of the hybrid system, particularly related to neural regeneration. The manuscript is organized and reads well. However, I have several major comments that need to be addressed to make the work re-considerable for publication:

1- Au nanoparticles are well-known for their neural regeneration capacity either alone or as coupled with chitosan. There are several similar or relevant studies in the literature such as: https://doi.org/10.1016/j.surneu.2008.01.057, https://journals.lww.com/jcma/Fulltext/2021/11000/Fabrication_of_hyaluronic_acid_gold_nanoparticles.6.aspx, https://doi.org/10.1016/j.bbrc.2019.03.189

Thus, at the first step, the authors should justify the novelty and significance of their study compared to the literature.
Answer:
Thanks for the valuable suggestion from the Reviewer. We have included the comparison of various literatures in the “Discussion” section. (Page 20, Line 663-670)
“…Previous research has proved that Chi-Au conduit can stimulate the expression of brain-derived neurotrophic factor (BDNF) and glial cell line-derived neurotrophic factor (GDNF) in NSCs for nerve tissue regeneration [65]. However, NSCs are difficult to harvest. Additionally, they also need bioactive molecules to support proliferation [66]. Adipose-derived stem cells (ASCs) have also been applied for nerve regeneration [67]. The limitations of ASCs are also existed, for example, lacking of self-renewal ability and risk of malignant transformation [68]. Therefore, MSCs are the promising cell resource for nerve repair…”
Reference:
65. Lin, Y.-L.; Jen, J.-C.; Hsu, S.-h.; Chiu, M. Sciatic nerve repair by microgrooved nerve conduits made of chitosan-gold nanocomposites. Surgical neurology 2008, 70, S9-S18.
66. Amariglio, N.; Hirshberg, A.; Scheithauer, B.W.; Cohen, Y.; Loewenthal, R.; Trakhtenbrot, L.; Paz, N.; Koren-Michowitz, M.; Waldman, D.; Leider-Trejo, L. Donor-derived brain tumor following neural stem cell transplantation in an ataxia telangiectasia patient. PLoS medicine 2009, 6, e1000029.
67. Razavi, S.; Seyedebrahimi, R.; Jahromi, M. Biodelivery of nerve growth factor and gold nanoparticles encapsulated in chitosan nanoparticles for schwann-like cells differentiation of human adipose-derived stem cells. Biochemical and biophysical research communications 2019, 513, 681-687.
68. Hassanshahi, A.; Hassanshahi, M.; Khabbazi, S.; Hosseini‐Khah, Z.; Peymanfar, Y.; Ghalamkari, S.; Su, Y.W.; Xian, C.J. Adipose‐derived stem cells for wound healing. Journal of cellular physiology 2019, 234, 7903-7914.

2- Too long introduction with unnecessary explanation about stem cells.
Answer:
Thanks for the suggestion from the Reviewer. We have revised the explanation about stem cells in the “Introduction” section. (Page 2-3, Line 98-107)
“…The stem cell treatments used for clinical neurodegenerative diseases include embryonic stem cells (ES cells), neural stem cells (NSCs), and mesenchymal stem cells (MSCs) [32]. MSCs are abundant in the human body with the majority present in bone marrow, which can differentiate into various cell types with multifunctionality [33]. Furthermore, MSCs can also be obtained from adipose tissue [34], muscle [35], dental pulp [36] and umbilical blood [37]. According to previous research, MSCs induce lowest rate of immunological rejection in clinical applications [38,39]. MSCs have been demonstrated to be an efficient cell resource in therapies for neurological disorders [40]. The combination of MSCs and suitable biomedical substrates can strengthen tissue regeneration by secreting various types of growth factors and cytokines [41]…”
Reference:
32. Sakthiswary, R.; Raymond, A.A. Stem cell therapy in neurodegenerative diseases: From principles to practice. Neural regeneration research 2012, 7, 1822.
33. Samsonraj, R.M.; Raghunath, M.; Nurcombe, V.; Hui, J.H.; van Wijnen, A.J.; Cool, S.M. Concise review: multifaceted characterization of human mesenchymal stem cells for use in regenerative medicine. Stem cells translational medicine 2017, 6, 2173-2185.
34. Aust, L.; Devlin, B.; Foster, S.; Halvorsen, Y.; Hicok, K.d.; Du Laney, T.; Sen, A.; Willingmyre, G.; Gimble, J. Yield of human adipose-derived adult stem cells from liposuction aspirates. Cytotherapy 2004, 6, 7-14.
35. Jackson, W.M.; Nesti, L.J.; Tuan, R.S. Potential therapeutic applications of muscle-derived mesenchymal stem and progenitor cells. Expert opinion on biological therapy 2010, 10, 505-517.
36. Shi, S.; Gronthos, S. Perivascular niche of postnatal mesenchymal stem cells in human bone marrow and dental pulp. Journal of bone and mineral research 2003, 18, 696-704.
37. Corrao, S.; La Rocca, G.; Lo Iacono, M.; Corsello, T.; Farina, F.; Anzalone, R. Umbilical cord revisited: from Wharton’s jelly myofibroblasts to mesenchymal stem cells. 2013.
38. Han, Y.; Li, X.; Zhang, Y.; Han, Y.; Chang, F.; Ding, J. Mesenchymal stem cells for regenerative medicine. Cells 2019, 8, 886.
39. Börger, V.; Bremer, M.; Ferrer-Tur, R.; Gockeln, L.; Stambouli, O.; Becic, A.; Giebel, B. Mesenchymal stem/stromal cell-derived extracellular vesicles and their potential as novel immunomodulatory therapeutic agents. International Journal of Molecular Sciences 2017, 18, 1450.
40. Przyborski, S.A.; Hardy, S.A.; Maltman, D.J. Mesenchymal stem cells as mediators of neural differentiation. Current stem cell research & therapy 2008, 3, 43-52.
41. Boomsma, R.A.; Geenen, D.L. Mesenchymal stem cells secrete multiple cytokines that promote angiogenesis and have contrasting effects on chemotaxis and apoptosis. PloS one 2012, 7, e35685.

3- Page 3, line 124, why osteogenesis of MSCs should matter in this study dealing with neural tissue regeneration?
Answer:
Thanks for the suggestion from the reviewer. We have deleted the description about osteogenesis in the last paragraph of “Introduction” section.

4- Page 4, line 180; how much was the electron accelerating voltage? any sputter coating?
Answer:
Thanks for the comment from the reviewer. The solution of Au nanoparticles was firstly added on a silicon wafer and dried out at 80°C. The observation was set at 5.0 kV. We have also modified the description for SEM in section 2.2.5.
“…Briefly, 10 μL of Au nanoparticle solution was added on a silicon wafer and dried out at 80°C. After dried out, the silicon wafer with Au nanoparticles was sputter coated with silver. Next, the voltage was set at 5.0 kV to observe the shape of Au nanoparticles. The scale bar was set at 1 μm….” (Page 4, Line 175-178)

5- Page 7, line 329-334; Au chelation should lead to disappearance of characteristic peaks of chitosan. The observed shifts can be mainly associated to H-bonding. Why it is so? describe the mechanism? 
Answer:
Thanks for the comment from the Reviewer. We have included the new description for the covalent bonding of Chi-Au nanoparticles in the “Discussion” section. (Page 19-20, Line 634-642)
“Based on the FTIR results in present study (Figure 1D), the absorption peak at around 3600-3400 cm-1 represented as O-H stretching bond, which plays a vital role for the stability of Chi-Au nanoparticles in aqueous solution [53]. Additionally, the amide I (1640 cm-1, C=O stretching) and amide II (1540 cm-1, N-H bending) bands [54] are explored to be significantly changed after the chelation of Chi and Au nanoparticles (Figure 1D). The covalent bonding of amide I group represents carboxylic group, which is shifted due to the linkage of Chi and Au nanoparticles [55]. Simultaneously, the shifted peak of N-H bending indicates the bonding of Au to nitrogen atoms [55,56]. The results verify that Au nanoparticles are coupled with chitosan.”
Reference:
53. Jiang, C.; Zhu, J.; Li, Z.; Luo, J.; Wang, J.; Sun, Y. Chitosan–gold nanoparticles as peroxidase mimic and their application in glucose detection in serum. RSC advances 2017, 7, 44463-44469.
54. Silva, I.O.; Ladchumananandasivam, R.; Nascimento, J.H.O.; Silva, K.K.O.; Oliveira, F.R.; Souto, A.P.; Felgueiras, H.P.; Zille, A. Multifunctional chitosan/gold nanoparticles coatings for biomedical textiles. Nanomaterials 2019, 9, 1064.
55. Komalam, A.; Muraleegharan, L.G.; Subburaj, S.; Suseela, S.; Babu, A.; George, S. Designed plasmonic nanocatalysts for the reduction of eosin Y: absorption and fluorescence study. International Nano Letters 2012, 2, 1-9.
56. Primo, A.; Quignard, F. Chitosan as efficient porous support for dispersion of highly active gold nanoparticles: design of hybrid catalyst for carbon–carbon bond formation. Chemical communications 2010, 46, 5593-5595.

6- line 339; why the reported values are so detailed? They should be rounded up/down.
Answer:
Thanks for the valuable comment from the Reviewer. We have modified the description in Section 3.2 and Figure S2 caption in supplementary data.
“Based on the results, the expression of CD14, CD34, and CD45 markers were quantified as 1% (negative markers), which indicated MSCs did not express endothelial differentiation markers. Additionally, large amounts of cells expressed CD44 (99%), CD90 (97%), and CD105 (96%) (positive markers)” (Page 8, Line 330-333)

7- The larger colonization of MSCs on the nanocomposite with higher density of Au nanoparticles could be related to enhanced electrical conductivity of the system that allows cell-cell communication. Can the authors comment on this aspect of their system?
Answer:
Thanks for the suggestion from the Reviewer. We have also included the description in the “Discussion” section. (Page 20, Line 660-663)
“In addition, the superior physicochemical properties of Au (~ 5 nm) can facilitate the electrical conductivity and strengthen the cell-cell interaction [64], which is corresponded to the results of increased MSCs colonization in present study (Figure 2A & 2B).”
Reference:
64. Kumar, P.P.P.; Lim, D.-K. Gold-polymer nanocomposites for future therapeutic and tissue engineering applications. Pharmaceutics 2021, 14, 70.

8- Page 11, line 466; increased level of MMP-9 could be indicative of intensive inflammatory response, as well. Correct?
Answer:
Thanks for the comment from the reviewer. The new description for MMP-9 expression was included in Discussion. (Page 20, Line 684-688)
“The matrix metalloproteinase could cause the remodeling of ECM for cell migration. The higher expression level of MMP-9 proteins was verified to facilitate NSPCs to differentiate into migratory cells for neural regeneration [74]. The results shown in Figure 4A & 4B indicate Chi-Au nanocomposites induced MMP-9 expression in MSCs, particularly at a concentration of 50 ppm Au.”
Reference
:
74. Barkho, B.Z.; Munoz, A.E.; Li, X.; Li, L.; Cunningham, L.A.; Zhao, X. Endogenous matrix metalloproteinase (MMP)‐3 and MMP‐9 promote the differentiation and migration of adult neural progenitor cells in response to chemokines. Stem Cells 2008, 26, 3139-3149.

9- Page 14, line 507; the presentation of the results in the text format is confusing. A table or visual element that can summarize and present the data would be more appealing. 
Answer:
Thanks for the valuable comment from the Reviewer. We have format the text presentation of mRNA expression results into Tables.
(Page 13, Section 3.6.)
Table 1: GFAP mRNA expression at 3, 5 and 7 days.
Table 2: β-Tubulin mRNA expression at 3, 5 and 7 days.
Table 3: Nestin mRNA expression at 3, 5 and 7 days.
Table 4: CD31 mRNA expression at 3, 5 and 7 days.
Table 5: vWF mRNA expression at 3, 5 and 7 days.

10- The optimized biological performance of Chi-Au 50 ppm over that with higher ppm of Au nanoparticles could be attributed to which factors?
Answer:
Thanks for the comment from the Reviewer. We have discussed the issue and included in Discussion. (Page 20, Line 657-660)
“The microstructure of nanocomposite can be improved by low amount of Au, leading to the favorable biological performance. However, the high concentration of Au nanoparticles (100 ppm) may cause adverse impact on the surface morphology of nanocomposite due to the aggregation [62,63].”
Reference:
62. Hung, H.S.; Chu, M.Y.; Lin, C.H.; Wu, C.C.; Hsu, S.h. Mediation of the migration of endothelial cells and fibroblasts on polyurethane nanocomposites by the activation of integrin‐focal adhesion kinase signaling. Journal of Biomedical Materials Research Part A 2012, 100, 26-37.
63. Aldrich, S. Gold nanoparticles: properties and applications. Sigma-Aldrich, St. Louis, MO 2015.

11- Figure 1Ba, SEM image is in poor quality, scale bar is hardly read and the dots are not clear what they represent?
Answer:
Thanks for the valuable suggestion from the reviewer. We have revised the SEM image for Au nanoparticles with better quality. (Page 7, Figure 1B-a)

12- Discussion; the statements about peripheral nerve injuries are unnecessary.  In general, discussion part does not analyze and discuss the achieved results rather repeats some general information about importance of nerve regeneration using nanomaterials, particularly Au nanoparticles, etc. The paper mainly reports the results and not scientifically discusses the reason behind the observed behaviours and the findings.
Answer:
Thanks for the valuable suggestion from the reviewer.
(1) We have removed the description about peripheral nerve injuries in the first paragraph of Discussion section. The beginning of first paragraph is described as below: (Page 19, Line 611-613)
“The human nerve growth rate can reach approximately 2 – 5 mm per day [45], which was considered to be slow for nerve regeneration. Therefore, there is a need to develop a highly safe and effective drug to enhance nerve growth in clinical treatments….”
(2) We have discussed the reason behind the observed findings and included in the “Discussion” section.
“Based on the FTIR results in present study (Figure 1D), the absorption peak at around 3600-3400 cm-1 represented as O-H stretching bond, which plays a vital role for the stability of Chi-Au nanoparticles in aqueous solution [53]. Additionally, the amide I (1640 cm-1, C=O stretching) and amide II (1540 cm-1, N-H bending) bands [54] are explored to be significantly changed after the chelation of Chi and Au nanoparticles (Figure 1D). The covalent bonding of amide I group represents carboxylic group, which is shifted due to the linkage of Chi and Au nanoparticles [55]. Simultaneously, the shifted peak of N-H bending indicates the bonding of Au to nitrogen atoms [55,56]. The results verify that Au nanoparticles are coupled with chitosan.”
(Page 19-20, Line 634-642)
“The microstructure of nanocomposite can be improved by low amount of Au, leading to the favorable biological performance. However, the high concentration of Au nanoparticles (100 ppm) may cause adverse impact on the surface morphology of nanocomposite due to the aggregation [62,63]. In addition, the superior physicochemical properties of Au (~ 5 nm) can facilitate the electrical conductivity and strengthen the cell-cell interaction [64], which is corresponded to the results of increased MSCs colonization in present study (Figure 2A & 2B). Previous research has proved that Chi-Au conduit can stimulate the expression of brain-derived neurotrophic factor (BDNF) and glial cell line-derived neurotrophic factor (GDNF) in NSCs for nerve tissue regeneration [65]. However, NSCs are difficult to harvest. Additionally, they also need bioactive molecules to support proliferation [66]. Adipose-derived stem cells (ASCs) have also been applied for nerve regeneration [67]. The limitations of ASCs are also existed, for example, lacking of self-renewal ability and risk of malignant transformation [68]. Therefore, MSCs are the promising cell resource for nerve repair. “
(Page 20, Line 657-670)
“Nestin is type VI intermediate filament protein which expressed in nerve cells for the radial growth of axon. Nestin is also expressed in the development of CNS [69]. Furthermore, nestin is found to express in NSPCs and astroglia cells [70]. GFAP is the glial fibrillary acidic protein, which is the marker of glial cells. GFAP was found to increase expression while stem cells differentiate into glial cells [71]. The microtubules are the cytoskeleton composed of α and β-tubulin, which is associated with cell migration of newly developed neurons [72]. MSCs is the cell resource with multiple differentiation capacities. Therefore, the investigation of nestin, GFAP and β-tubulin expression in MSCs plays vital role during neural differentiation when culturing with Chi-Au nanocomposites.” (Page 20, Line 671-679)
“The matrix metalloproteinase could cause the remodeling of ECM for cell migration. The higher expression level of MMP-9 proteins was verified to facilitate NSPCs to differentiate into migratory cells for neural regeneration [74]. The results shown in Figure 4A & 4B indicate Chi-Au nanocomposites induced MMP-9 expression in MSCs, particularly at a concentration of 50 ppm Au. Moreover, based on the FACS result exhibited in Figure 2C, the expression of CD44 in MSCs was found to significantly lower in the Chi-Au 50 ppm treatment. Previous research has elucidated that the expression of β-tubulin neuronal markers in bone-marrow MSCs increased while the CD44 expression decreased [75]. Our results confirmed that Chi-Au 50 ppm could facilitate MSCs undergo differentiation.” (Page 20, Line 684-692)

Round 2

Reviewer 1 Report

Dear Authors

the paper has been improved according to the referee's instructions, so I believe the document could be accepted with a minor revision.

Minor revision:

- line 374-375; delete the sentence "while the scale bar for CD44 immunostaining was 50 μm.

-line 375-376. delete the sentence "Cell nuclei were located by DAPI (blue)".

- delete in Figure 9 and relative legend every reference to maintaining stemness properties.

- please review all the legends of the modified figures.

Best regards

Author Response

Reviewer 1:

The paper has been improved according to the referee's instructions, so I believe the document could be accepted with a minor revision.

Minor revision:

  1. line 374-375; delete the sentence "while the scale bar for CD44 immunostaining was 50 μm.
    Answer:
    Thanks for the suggestion from the reviewer. We have deleted the sentence. (Page 9, Line 367-375)
    Figure 2. Colony formation and CD44 expression of MSCs cultured on Chi-Au nanocomposites. (A) Colony formation ability and (B) colony formation number of MSCs were observed at 1, 3, 5, and 7 days. Furthermore, (C) the expression of CD44 marker at Day 7 was semi-quantified by fluorescence activated cell sorting (FACS) method. The results demonstrated Chi combined with Au could induce proliferation capacities of MSCs, particularly at the concentration 50 and 100 ppm of Au nanoparticles. Additionally, the CD44 expression was explored to be significantly lower in Chi-Au 25 and 50 ppm group. The scale bar to observe colony formation was 100 μm. CD44 marker was conjugated with FITC (green). Data are exhibited as the mean ± SD (n = 3). *p < 0.05, **p < 0.01, ***p < 0.001: compared to the Control (TCPS).”
  2. line 375-376. delete the sentence "Cell nuclei were located by DAPI (blue)".
    Answer:
    Thanks for the suggestion from the reviewer. We have deleted the sentence. (Page 9, Line 367-375)
    Figure 2. Colony formation and CD44 expression of MSCs cultured on Chi-Au nanocomposites. (A) Colony formation ability and (B) colony formation number of MSCs were observed at 1, 3, 5, and 7 days. Furthermore, (C) the expression of CD44 marker at Day 7 was semi-quantified by fluorescence activated cell sorting (FACS) method. The results demonstrated Chi combined with Au could induce proliferation capacities of MSCs, particularly at the concentration 50 and 100 ppm of Au nanoparticles. Additionally, the CD44 expression was explored to be significantly lower in Chi-Au 25 and 50 ppm group. The scale bar to observe colony formation was 100 μm. CD44 marker was conjugated with FITC (green). Data are exhibited as the mean ± SD (n = 3). *p < 0.05, **p < 0.01, ***p < 0.001: compared to the Control (TCPS).”
  3. delete in Figure 9 and relative legend every reference to maintaining stemness properties.
    Answer:
    Thanks for the valuable comment from the reviewer. We have modified the illustration in Figure 9 associated with the description of maintaining stemness properties, and the Figure 9 legend as well. (Page 19, Line 607-610)

    Figure 9. Summary illustrations for Chi-Au nanocomposites in in vitro and in vivo assessments. The summary shows Chi combined with 50 ppm of Au nanoparticles can significantly induce neural differentiation. Furthermore, the animal model also indicated Chi-Au 50 ppm was a highly safe nanocomposite, indicating its potential for use in tissue engineering.”

5. please review all the legends of the modified figures
Answer:
Thanks for the valuable suggestion from the reviewer. We have confirmed all the legends of the modified figures.

Reviewer 2 Report

Dear Editor,

The manuscript still needs significant improvement. I think the novelty and significance of the works have not been properly and confidently justified. The discussion part is week and inconsistent. FTIR results show only shift of characteristic peaks which not necessarily means chelation rather it could be simply a hydrogen bonding. If the latter would be the case, what functional groups on Au are playing a role? Why at the highest Au concentration, agglomeration takes place? The tabulation of data could be applied to other sections/results after section 3.6 as well. Figure 1Ba is still low quality. what does this statement mean: “The microstructure of nanocomposite can be improved by low amount of Au, leading to the favorable biological performance"? what kind of microstructure? How it affects the biological performance? My comment 8 concerning inflammatory response has not been answered.

Author Response

Reviewer 2:

The manuscript still needs significant improvement. I think the novelty and significance of the works have not been properly and confidently justified.

  1. The discussion part is weak and inconsistent.
    Answer:
    Thanks the valuable comment from the reviewer. We have reorganized the “Discussion” section to further correspond to the “Results” section.
    (1) “Au nanoparticles have been shown to have potential in various medical fields, including tissue engineering and regenerative medicine [50]. Au also possesses unique physicochemical properties, such as low cytotoxicity, and it can be synthesized easily [51]. Based on the FTIR results in present study (Figure 1D), the absorption peak at around 3600-3400 cm-1 represented as O-H stretching bond, which plays a vital role for the stability of Chi-Au nanoparticles in aqueous solution [52]. Additionally, the amide I (1640 cm-1, C=O stretching) and amide II (1540 cm-1, N-H bending) bands [53] are explored to be significantly changed after the combination of Chi and Au nanoparticles (Figure 1D). The covalent bonding of amide I group represents carboxylic group, which is shifted due to the linkage of Chi and Au nanoparticles [54]. Simultaneously, the shifted peak of N-H bending indicates the bonding of Au to nitrogen atoms [54,55]. The changes from amide bands were proved owing to the electrostatic interactions between the particle surface of Au and amide groups [56]. The results verify that Au nanoparticles are coupled with chitosan in the present research. Furthermore, Au nanoparticles can modify the surface topography of natural and synthetic polymers, conferring favorable properties at a low concentration owing to their extremely small size, i.e., at the nano-scale (5 ~ 50 nm) [57], leading to the improvement of surface physicochemical properties. The surface roughness would influence the cell aggregation owing to the interaction between nanocomposites and cells [58,59]. According to our AFM images (Figure S2B), the RMS values of Chi-Au 50 and 100 ppm nanocomposites were 0.67 and 0.38 nm, indicating the uniform surface topography for cell aggregation. In addition, the superior physicochemical properties of Au (~ 5 nm) can facilitate the electrical conductivity and strengthen the cell-cell interaction [60], which is corresponded to the results of enhancing MSCs proliferation and differentiation capacity in the present study (Figure 2A & 2B).” (Page 20-21, Line 631-654)
    (2) “The versatile applications of Au nanoparticles can create appropriate nanocomposite scaffolds to enhance cell viability and differentiation capacities [61]. A recent study demonstrated that a chitosan/κ-carrageenan hydrogel modified with Au nanoparticles strengthened the viability and attachment of MD-63 cells [62]. A major goal in the development of biomaterials is to enhance the differentiation capacity of stem cells using biocompatible Au-modified nanocomposites. For example, embryonic stem cells (ESCs) and mesenchymal stem cells (MSCs) possess superior self-renewal abilities, and can differentiate into various cell types (neurons and endothelial cells) through the induction of cell signals [63]. It has been shown that Au nanoparticles could facilitate the differentiation of ESCs into dopaminergic (DA) neurons, and an association has been shown between the mTOR/p70S6K signaling pathway and differentiation of dopaminergic (DA) neurons [64]. Furthermore, FGF2-mediated and RA (retinoic acid)-mediated differentiated pathways were also found to be related to the neuronal differentiation of ESCs induced by Au-modified polymers [65]. Previous research has proved that Chi-Au conduit can stimulate the expression of brain-derived neurotrophic factor (BDNF) and glial cell line-derived neurotrophic factor (GDNF) in NSCs for nerve tissue regeneration [66]. However, NSCs are difficult to harvest. Additionally, they also need bioactive molecules to support proliferation [67]. Adipose-derived stem cells (ASCs) have also been applied for nerve regeneration [68]. The limitations of ASCs are also existed, for example, lacking of self-renewal ability and risk of malignant transformation [69]. Therefore, MSCs are the promising cell resource for nerve repair.” (Page 21, Line 655-675)

  2. FTIR results show only shift of characteristic peaks which not necessarily means chelation rather it could be simply a hydrogen bonding. If the latter would be the case, what functional groups on Au are playing a role?
    Answer:
    Thanks the valuable comment from the reviewer.We have included the description for the bonding of Au and chitosan polymer in the “Discussion” section.
    “Additionally, the amide I (1640 cm-1, C=O stretching) and amide II (1540 cm-1, N-H bending) bands [53] are explored to be significantly changed after the combination of Chi and Au nanoparticles (Figure 1D). The covalent bonding of amide I group represents carboxylic group, which is shifted due to the linkage of Chi and Au nanoparticles [54]. Simultaneously, the shifted peak of N-H bending indicates the bonding of Au to nitrogen atoms [54,55]. The changes from amide bands were proved owing to the electrostatic interactions between the particle surface of Au and amide groups [56]. The results verify that Au nanoparticles are coupled with chitosan in the present research.” (Page 20-21, Line 636-644)

  3. Why at the highest Au concentration, agglomeration takes place?
    Answer:
    (1) Based on the previous research, the appropriate amount of metal nanoparticles can lead to homogeneous suspension on polymer substrate. However, the high concentration of Au nanoparticles will agglomerate due to the plasmon resonance between metal to metal [1]. Moreover, the nanocomposites with high concentration of Au will lead to Au aggregation, which is due to the destroy of hydrogen-bonded carbonyl band [2].
    Reference:
    1. Cárdenas-Triviño, G.; Cruzat-Contreras, C. Study of aggregation of Gold Nanoparticles in Chitosan. Journal of Cluster Science 2018, 29, 1081-1088.
  4. Hsu, S.-h.; Tang, C.-M.; Tseng, H.-J. Gold nanoparticles induce surface morphological transformation in polyurethane and affect the cellular response. Biomacromolecules 2008, 9, 241-248.
    (2) We also have deleted the description related to Au aggregation in the “Discussion” section to further correspond to the AFM results.

  5. The tabulation of data could be applied to other sections/results after section 3.6 as well.
    Answer:
    Thanks for the comment from the reviewer. We have revised the subtitles in the “Results” section. The new section 3.6. has applied to described the mRNA expression investigated by real-time PCR assay. And the tabulations of the data were moved to Page 14. Furthermore, the subtitle for FACS and IF measurement was described for section 3.7. The in vivo results were moved to section 3.8.
    The subtiltles were described as followed:
    “3.6. Real-Time PCR assay for mRNA expression induced by Chi-Au” (Page 13, Line 458)
    “3.6.1. Tabulation data of mRNA expression induced by Chi-Au nanocomposites” (Page 14, Line 476)
    “3.7. FACS and IF measurement for protein expression induced by Chi-Au” (Page 15, Line 487)
    “3.8. Subcutaneous implantation in an animal model for biocompatibility measurements” (Page 16, Line 544)

  6. Figure 1Ba is still low quality.
    Answer:
    Thanks for the comment from the reviewer. The new SEM image of Au nanoparticles was replaced in the Figure 1B-a. (Page 7)

  7. what does this statement mean: “The microstructure of nanocomposite can be improved by low amount of Au, leading to the favorable biological performance"? what kind of microstructure? How it affects the biological performance?
    Answer:
    We have reworded the description in the “Discussion” section. (Page 21, Line 645-655)
    “Furthermore, Au nanoparticles can modify the surface topography of natural and synthetic polymers, conferring favorable properties at a low concentration owing to their extremely small size, i.e., at the nano-scale (5 ~ 50 nm) [57], leading to the improvement of surface physicochemical properties. The surface roughness would influence the cell aggregation owing to the interaction between nanocomposites and cells [58,59]. According to our AFM images (Figure S2B), the RMS values of Chi-Au 50 and 100 ppm nanocomposites were 0.67 and 0.38 nm, indicating the uniform surface topography for cell aggregation. In addition, the superior physicochemical properties of Au (~ 5 nm) can facilitate the electrical conductivity and strengthen the cell-cell interaction [60], which is corresponded to the results of enhancing MSCs proliferation and differentiation capacity in the present study (Figure 2A & 2B).”

  8. My comment 8 concerning inflammatory response has not been answered.
    Answer:
    MMP-9 is the enzyme associated with the proteolysis of extracellular matrix (ECM) [1]. The MMP-9 can maintain the normal physiological function such as angiogenesis [2] and embryogenesis [3]. The increased of MMP-9 secretion is also associated with inflammatory response and tumor progression [4]. The expression of MMP-9 cam be regulated by cytokines, such as tumor necrosis factor-α (TNF-α) [1]. However, the expression of MMP-9 has been proved to express in proliferating and differentiated cells during neurogenesis [5] including the brain vascularization and neurite extension related to ECM remodelling [6]. Our previous research also indicated the high expression of MMP-9 could enhance stem cells migration and differentiation for neural tissue regeneration [7]. Therefore, the results of higher MMP-9 expression induced by Chi-Au 50 ppm nanocomposites in the present research could be supported.
    Reference:
    1. Chambaut‐Guérin, A.M.; Hérigault, S.; Rouet‐Benzineb, P.; Rouher, C.; Lafuma, C. Induction of matrix metalloproteinase MMP‐9 (92‐kDa gelatinase) by retinoic acid in human neuroblastoma SKNBE cells: relevance to neuronal differentiation. Journal of neurochemistry 2000, 74, 508-517.
    2. Roy, R.; Zhang, B.; Moses, M.A. Making the cut: protease-mediated regulation of angiogenesis. Experimental cell research 2006, 312, 608-622.
    3. Vu, T.H.; Werb, Z. Matrix metalloproteinases: effectors of development and normal physiology. Genes & development 2000, 14, 2123-2133.
    4. Deryugina, E.I.; Quigley, J.P. Matrix metalloproteinases and tumor metastasis. Cancer and metastasis reviews 2006, 25, 9-34.
    5. Gao, W.-Q.; Heintz, N.; Hatten, M.E. Cerebellar granule cell neurogenesis is regulated by cell-cell interactions in vitro. Neuron 1991, 6, 705-715.
    6. Soler, R.C.; Gui, Y.-H.; Linask, K.K.; Muschel, R.J. MMP-9 (gelatinase B) mRNA is expressed during mouse neurogenesis and may be associated with vascularization. Developmental Brain Research 1995, 88, 37-52.
    7. Yang, M.-Y.; Liu, B.-S.; Huang, H.-Y.; Yang, Y.-C.; Chang, K.-B.; Kuo, P.-Y.; Deng, Y.-H.; Tang, C.-M.; Hsieh, H.-H.; Hung, H.-S. Engineered Pullulan-Collagen-Gold Nano Composite Improves Mesenchymal Stem Cells Neural Differentiation and Inflammatory Regulation. Cells 2021, 10, 3276.

Round 3

Reviewer 2 Report

Dear editor,

Taking into account my suggested corrections, the revised paper is now publishable.